# Improving Adversarial Robust Fairness via Anti-Bias Soft Label Distillation

**Shiji Zhao**[1], **Ranjie Duan**[2], **Xizhe Wang**[1], **Xingxing Wei**[1][*]

[1]Institute of Artificial Intelligence, Beihang University, Beijing, China
[2]Security Department, Alibaba Group, Hangzhou, China
{zhaoshiji123,xizhewang,xxwei}@buaa.edu.cn, ranjieduan@gmail.com

## Abstract

Adversarial Training (AT) has been widely proved to be an effective method to improve the adversarial robustness against adversarial examples for Deep Neural Networks (DNNs). As a variant of AT, Adversarial Robustness Distillation (ARD) has demonstrated its superior performance in improving the robustness of small student models with the guidance of large teacher models. However, both AT and ARD encounter the robust fairness problem: these models exhibit strong robustness when facing part of classes (easy class), but weak robustness when facing others (hard class). In this paper, we give an in-depth analysis of the potential factors and argue that the smoothness degree of samples' soft labels for different classes (i.e., hard class or easy class) will affect the robust fairness of DNNs from both empirical observation and theoretical analysis. Based on the above finding, we propose an Anti-Bias Soft Label Distillation (ABSLD) method to mitigate the adversarial robust fairness problem within the framework of Knowledge Distillation (KD). Specifically, ABSLD adaptively reduces the student's error risk gap between different classes to achieve fairness by adjusting the class-wise smoothness degree of samples' soft labels during the training process, and the smoothness degree of soft labels is controlled by assigning different temperatures in KD to different classes. Extensive experiments demonstrate that ABSLD outperforms state-of-the-art AT, ARD, and robust fairness methods in the comprehensive metric (Normalized Standard Deviation) of robustness and fairness.

## 1 Introduction

Deep neural networks (DNNs) have achieved great success in various tasks, e.g., classification [10], detection [6], and segmentation [24]. However, DNNs are vulnerable to adversarial attacks [30; 35; 33; 34], where adding small perturbations to the input examples will lead to misclassification. To enhance the robustness of DNNs, Adversarial Training (AT) [20; 41; 32; 14] is proposed and has been proven to be an effective method to defend against adversarial examples. To further improve the robustness, Adversarial Robustness Distillation (ARD) [7] as a variant of AT is proposed and aims to transfer the robustness of the large models into the small models based on Knowledge Distillation (KD), and further researches [44; 45; 43; 12; 42] show the excellent performance of ARD.

Although AT and ARD can remarkably improve the adversarial robustness, some researches [2; 31; 39; 19; 28; 38] demonstrate the robust fairness problem: these models perform strong robustness on part of classes (easy class) but show high vulnerability on others (hard class). This phenomenon will raise further attention to class-wise security. Specifically, an overall robust model appears to be relatively safe for model users, however, the robust model with poor robust fairness will lead to attackers targeting vulnerable classes of the model, which leads to significant security risks to

---

[*]Corresponding Author.

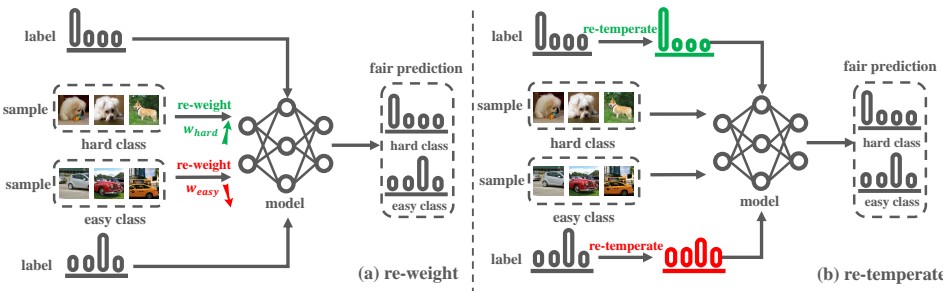

Figure 1: The comparison between the sample-based fair adversarial training and our label-based fair adversarial training. For the former ideology in (a), the trained model's bias is avoided by re-weighting the sample's importance according to the different contribution to fairness. For the latter ideology in (b), the trained model's bias is avoided by re-temperating the smoothness degree of soft labels for different classes.

potential applications. Different from simply improving the overall robustness, some methods are proposed to address the robust fairness problem in AT and ARD [39; 36; 38; 19; 28] (i.e., improving the worst-class robustness as much as possible without sacrificing too much overall robustness). However, the robust fairness problem still exists and requires further to be explored.

For that, we give an in-depth analysis of the potential factors to influence robust fairness in the optimization objective function. From the perspective of the training sample, the sample itself has a certain degree of biased behavior, which is mainly reflected in the different learning difficulties and various vulnerabilities to adversarial attacks. For this reason, previous works apply the re-weighting ideology to achieve robust fairness for different types of classes in the optimization process [39; 36; 38]. However, as another important factor in the optimization objective function, the role of the labels applied to guide the model's training is ignored. Labels can be divided into two types, including one-hot labels and soft labels, where the soft labels are widely studied [29; 11] and have been proven effective in improving the performance of DNNs. Inspired by this, we try to explore robust fairness from the perspective of samples' soft labels. Interestingly, we first find that the smoothness degree of soft labels for different classes (i.e., hard and easy class) can affect the robust fairness of DNNs from both empirical observation and theoretical analysis. Intuitively speaking, sharper soft labels mean larger supervision intensity, while smoother soft labels mean smaller supervision intensity, so it is helpful to improve robust fairness by assigning sharp soft labels for hard classes and smooth soft labels for easy classes.

Based on the above finding, we further propose an Anti-Bias Soft Label Distillation (ABSLD) method to mitigate the adversarial robust fairness problem within the framework of knowledge distillation. ABSLD can adaptively adjust the smoothness degree of soft labels by re-temperating the teacher's soft labels for different classes, and each class has its own teacher's temperatures based on the student's error risk. For instance, when the student performs more error risk in some classes, ABSLD will compute sharp soft labels by assigning lower temperatures, and the student's learning intensity for these classes will relatively increase compared with other classes. After the optimization, the student's robust error risk gap between different classes will be reduced. The code can be found in `https://github.com/zhaoshiji123/ABSLD`.

Our contribution can be summarized as follows:

- We explore the labels' effects on the adversarial robust fairness of DNNs, which is different from the existing sample perspective. To the best of our knowledge, we are the first one to find that the smoothness degree of samples' soft labels for different types of classes can affect the robust fairness from both empirical observation and theoretical analysis.

- We propose the Anti-Bias Soft Label Distillation (ABSLD) to enhance the adversarial robust fairness within the framework of knowledge distillation. Specifically, we re-temperate the teacher's soft labels to adjust the class-wise smoothness degree and further reduce the student's error risk gap between different classes in the training process.

- We empirically verify the effectiveness of ABSLD. Extensive experiments on different datasets and models demonstrate that our ABSLD can outperform state-of-the-art methods against a variety of attacks in the comprehensive metric (Normalized Standard Deviation) of robustness and fairness.

## 2 Related Work

### 2.1 Adversarial Training

To defend against the adversarial examples, Adversarial Training (AT) [20; 41; 32; 13; 25] is regarded as an effective method to obtain robust models. AT can be formulated as a min-max optimization problem as follows:

$$\min_{\theta} E_{(x,y)\sim\mathcal{D}}[\max_{\delta\in\Omega} \mathcal{L}(f(x+\delta;\theta),y)],\tag{1}$$

where $f(\cdot;\theta)$ represents a deep neural network with weight $\theta$, $D$ represents a data distribution with clean example $x$ and the ground truth label $y$. $\mathcal{L}$ represents the optimization loss function, e.g. the cross-entropy loss. $\delta$ represents the adversarial perturbation, and $\Omega$ represents a bound, which can be defined as $\Omega = \{\delta : ||\delta|| \leq \epsilon\}$ with the maximum perturbation scale $\epsilon$. To further improve the performance, some variant methods of AT appear including regularization [21; 41; 32], using additional data [27; 22], and optimizing iteration process [15; 23]. Different from the above methods for improving the overall robustness, in this paper, we focus on solving the robust fairness problem.

### 2.2 Adversarial Robustness Distillation

Knowledge Distillation [11] as a training method can effectively transfer the large model's knowledge into the small model's knowledge, which has been widely applied in different areas. To enhance the adversarial robustness of small DNNs, Goldblum et al. [7] first propose Adversarial Robustness Distillation (ARD) by applying the clean prediction distribution of strong robust teacher models to guide the adversarial training of student models. Zhu et al. [44] argue that the prediction of the teacher model is not so reliable, and composite with unreliable teacher guidance and student introspection during the training process. RSLAD [45] applies the teacher clean prediction distribution as the guidance to train both clean examples and adversarial examples. MTARD [43; 42] applies clean teacher and adversarial teacher to enhance both accuracy and robustness, respectively. AdaAD [12] adaptively searches for optimal match points by directly applying the teacher adversarial prediction distribution in the inner maximization. In this paper, we explore how to enhance robust fairness within the framework of knowledge distillation.

### 2.3 Adversarial Robust Fairness

Some researchers address the robust fairness problem from different views [39; 18; 36; 38; 19; 37; 28] and improve the fairness without losing too much robustness. The most intuitive idea is to give different weights to the sample of different classes in the optimization process, and Xu et al. [39] propose Fair Robust Learning (FRL), which adjusts the loss weight and the adversarial margin based on the prediction accuracy of different classes. Ma et al. [19] finds the trade-off exists between robustness and fairness and propose Fairly Adversarial Training to mitigate this phenomenon by adding a regularization loss to control the variance of class-wise adversarial error risk. Sun et al. [28] propose Balance Adversarial Training (BAT) to achieve both source-class fairness (different difficulties in generating adversarial examples from each class) and target-class fairness (disparate target class tendencies when generating adversarial examples). Wu et al. [37] argue that the maximum entropy regularization for the model's prediction distribution can help to achieve robust fairness. Wei et al. [36] propose Class-wise Calibrated Fair Adversarial Training (CFA) to address fairness by dynamically customizing adversarial configurations for different classes and modifying the weight averaging operation. To enhance the ARD robust fairness, Yue et al. [38] propose Fair-ARD by re-weighting different classes based on the vulnerable degree. Different from these sample-based fairness methods, we try to solve this problem from the perspective of samples' labels, by adjusting the class-wise smoothness degree of samples' soft labels in the optimization process.

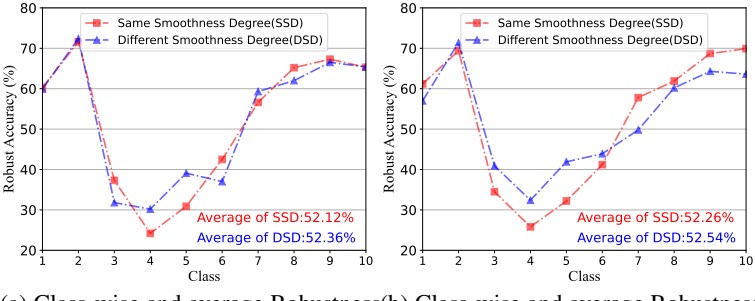

(a) Class-wise and average Robustness of ResNet-18 on CIFAR-10.    (b) Class-wise and average Robustness of MobileNet-v2 on CIFAR-10.

Figure 2: The class-wise and average robustness of DNNs guided by soft labels with the same smoothness degree (SSD) and different smoothness degree (DSD) for different classes, respectively. For the soft labels with different smoothness degrees, we use sharper soft labels for hard classes and use smoother soft labels for easy classes. We select two DNNs (ResNet-18 and MobileNet-v2) trained by SAT [20] on CIFAR-10. The robust accuracy is evaluated based on PGD. The checkpoint is selected based on the best checkpoint of the highest mean value of all-class average robustness and the worst class robustness following [36]. We see that blue lines and red lines have similar average robustness, but the worst robustness of blue lines are remarkably improved compared with red lines.

# 3 Robust Fairness via Smoothness Degree of Soft Labels

As an important part of the model optimization, label information used to guide the model plays an important role. The label can be divided into one-hot labels and soft labels, where one-hot labels only contain one class's information and soft labels can be considered as an effective way to alleviate over-fitting and improve the performance [29]. Previous methods usually ignore the class-wise smoothness degree of soft labels, either applying the same smoothness degree [29], or uniformly changing the smoothness degree of soft labels for all the classes without deliberate adjustments [11]. Different from previous methods, we are curious about *if we adjust the class-wise smoothness degree of soft labels, will it influence the class-wise robust fairness of the trained model?* Intuitively speaking, different smoothness degree of soft labels denote different supervision intensity, which means that it is possible to achieve fairness by adjusting the smoothness degree of soft labels. Here we try to explore the relationship between class-wise smoothness degree of soft labels and the robust fairness from both empirical observation and theoretical analysis.

## 3.1 Empirical Observation

Here, we focus on the impact of the class-wise smoothness degree of soft labels on adversarial training. First, we train the model with soft labels that have the same smoothness degree for all types of classes (smoothing coefficient[2] is 0.2). Then we assign different smoothness degrees of soft labels for hard classes and easy classes: specifically, we manually use sharper soft labels for hard classes (smoothing coefficient is 0.05) and smoother soft labels for easy classes (smoothing coefficient is 0.35). We conduct the experiment based on the SAT [20] shown in Figure 2.

The result shows that the class-wise smoothness degree of soft labels has an impact on class-wise robust fairness. When we apply the sharper smoothness degree of soft labels for hard classes and the smoother smoothness degree of soft labels for easy classes, the class-wise robust fairness problem can be alleviated. More specifically, for the two worst classes (class 4, 5), the robust accuracy of ResNet-18 guided by the soft label distribution with the same smoothness degree is 24.2%, and 30.9%, and the robust accuracy of ResNet-18 guided by the soft label distribution with different smoothness degree is 30.2%, and 39.1%, which exists an obvious improvement for the class-wise robust fairness, and the average robust accuracy has a slight improvement (52.12% vs 52.36%). Similar performance can also be observed in MobileNet-v2. This phenomenon indicates that appropriately assigning class-wise smoothness degrees of soft labels can be beneficial to achieve robust fairness.

---

[2]For N-class one-hot ground truth labels, after processing by the smoothing coefficient $\gamma$, the highest probability (correct class) decreases from 1 to $1 - \gamma$, and the other probability (wrong class) increases from 0 to $\frac{\gamma}{n-1}$.

## 3.2 Theoretical Analysis

Here we try to theoretically analyze the impact of the smoothness degree of soft labels on class-wise fairness. Firstly, we want to analyze the model bias performance with the guidance of the soft label distribution with the same smoothness degree. Then we give Corollary 1 by extending the prediction distribution of binary linear classifier into the prediction distribution of DNNs based on the theoretical analysis in [39] and [19].

**Corollary 1.** *A dataset $(x, y) \sim \mathcal{D}$ contains 2 classes (hard class $c_+$ and easy class $c_-$). Based on the label distribution $y$, the soft label distribution with same smoothness degree $P_{\lambda 1} = \{p_{c_+}^{\lambda 1}, p_{c_-}^{\lambda 1}\}$ can be generated and satisfies:*

$$1 > p_{c_-}^{\lambda 1}(x_{c_-}) = p_{c_+}^{\lambda 1}(x_{c_+}) > 0.5, \tag{2}$$

*If a DNN model $f$ is optimized by minimizing the average optimization error risk in $\mathcal{D}$ with the guidance of the equal soft labels $P_{\lambda 1} = \{p_{c_+}^{\lambda 1}, p_{c_-}^{\lambda 1}\}$, and obtain the relevant parameter $\theta_{\lambda 1}$, where the optimization error risk is measured by Kullback–Leibler divergence loss ($KL$):*

$$f(x; \theta_{\lambda 1}) = \arg \min_f \mathbb{E}_{(x,y) \sim \mathcal{D}}(KL(f(x; \theta_{\lambda 1}); P_{\lambda 1})), \tag{3}$$

*then the error risks (the expectation that samples are wrongly predicted by the model) for classes $c_+$ and $c_-$ have a relationship as follows:*

$$R(f(x_{c_+}; \theta_{\lambda 1})) > R(f(x_{c_-}; \theta_{\lambda 1})), \tag{4}$$

*where the error risks can be defined:*

$$R(f(x_{c_+}; \theta_{\lambda 1})) = \mathbb{E}_{(x,y) \sim \mathcal{D}}(CE(f(x_{c_+}; \theta_{\lambda 1}); y_{c_+})),$$
$$R(f(x_{c_-}; \theta_{\lambda 1})) = \mathbb{E}_{(x,y) \sim \mathcal{D}}(CE(f(x_{c_-}; \theta_{\lambda 1}); y_{c_-})). \tag{5}$$

Corollary 1 demonstrates that when optimizing hard and easy classes with equal intensity, the model will inevitably be biased, and this bias mainly comes from the characteristics of the sample itself and is not related to the optimization method. Based on the Corollary 1, we can further analyze the performance differences with the guidance of the different types of soft labels. Here we provide Theorem 1 about the relationship between class-wise smoothness degree of soft labels and fairness.

**Theorem 1.** *Following the setting in Corollary 1, for a dataset $\mathcal{D}$ containing 2 classes ($c_+$ and $c_-$), two soft label distribution ($P_{\lambda 1} = \{p_{c_+}^{\lambda 1}, p_{c_-}^{\lambda 1}\}$ and $P_{\lambda 2} = \{p_{c_+}^{\lambda 2}, p_{c_-}^{\lambda 2}\}$) exist, where $P_{\lambda 2}$ have a correct prediction distribution but have a limited different class-wise smoothness degree of soft labels ($v_1 > 0$, $v_2 > 0$):*

$$1 > p_{c_+}^{\lambda 2}(x_{c_+}) = p_{c_+}^{\lambda 1}(x_{c_+}) + v_1 > p_{c_+}^{\lambda 1}(x_{c_+}) =$$
$$p_{c_-}^{\lambda 1}(x_{c_-}) > p_{c_-}^{\lambda 2}(x_{c_-}) = p_{c_-}^{\lambda 1}(x_{c_-}) - v_2 > 0.5, \tag{6}$$

*then the model is trained with the guidance of the soft label distribution $P_{\lambda 1}$ and soft label distribution $P_{\lambda 2}$ and obtains the trained model parameters $\theta_{\lambda 1}$ and $\theta_{\lambda 2}$, respectively. If the model parameter $\theta_{\lambda 2}$ still satisfies: $R(f(x_{c_+}; \theta_{\lambda 2})) > R(f(x_{c_-}; \theta_{\lambda 2}))$, then the model's error risk for hard classes $c_+$ and easy classes $c_-$ has a relationship as follows:*

$$R(f(x_{c_+}; \theta_{\lambda 1})) - R(f(x_{c_-}; \theta_{\lambda 1})) > R(f(x_{c_+}; \theta_{\lambda 2})) - R(f(x_{c_-}; \theta_{\lambda 2})). \tag{7}$$

The proof of Theorem 1 can be found in Appendix A.1. Based on Theorem 1, the class-wise smoothness degree of soft labels theoretically has an impact on class-wise robust fairness. If the soft label distribution with different smoothness degree $P_{\lambda 2}$ is applied to guide the model training, where the sharper smoothness degree of soft labels for hard classes and smoother smoothness degree of soft labels for easy classes, the model will appear smaller error risk gap between easy and hard class compared with the soft label distribution with same smoothness degree $P_{\lambda 1}$, which demonstrates better robust fairness. The Theorem 1 theoretically demonstrates that if we appropriately adjust the class-wise smoothness degree of soft labels, the model can achieve class-wise robust fairness.

## 4 Anti-Bias Soft Label Distillation

### 4.1 Overall Framework

Based on the above finding, adjusting the class-wise smoothness degree of soft labels can be regarded as a potential way to obtain robust fairness. Then we consider introducing this ideology into Knowledge Distillation (KD), which has been proven to be an effective method to improve the robustness of small models [44; 45; 43; 12; 42]. Since the core idea of KD is to use the teacher's soft labels to guide the student's optimization process, we can adjust the class-wise smoothness degree of soft labels and obtain the student with both strong robustness and fairness.

Here we propose the Anti-Bias Soft Label Distillation (ABSLD) to obtain a student with adversarial robust fairness. We formulate the optimization objective function for ABSLD as follows:

$$\underset{f_s}{\arg\min} \, \mathbb{E}_{(x,y)\sim\mathcal{D}}(\mathcal{L}_{absld}(\tilde{x}, x; f_s, f_t^{'})), \tag{8}$$

$$s.t. \, R(f_s(\tilde{x}_k)) = \frac{1}{C}\sum_{i=1}^{C} R(f_s(\tilde{x}_i)), \tag{9}$$

where $x_i$ and $\tilde{x}_i$ are the clean examples and adversarial examples of the $i$-th class, $f_s$ denotes the student model, $f_t^{'}$ denotes the teacher model with Anti-Bias Soft Labels, C is the total number of classes, $\mathcal{L}_{absld}$ is the loss function, and $R(f_s(\tilde{x}_k))$ denotes the robust error risk of $k$-th class in student model $f_s$. Here we apply the Cross-Entropy loss $CE(f_s(\tilde{x}_k), y)$ as the evaluation criterion of the optimization error risk $R(f_s(\tilde{x}_k))$ following [19].

### 4.2 Re-temperate Teacher's Soft Labels

In order to adjust the class-wise smoothness degree of soft labels in ARD, two options exist: one is to use student feedback to update the teacher parameter in the process of optimizing students, but this option requires retraining the teacher model, which may bring the pretty optimization difficulty and computational overhead; the other is to directly adjust the smoothness degree of soft labels for different classes. Inspired by [42], we apply the temperature as a means of directly controlling the smoothness degree of soft labels during the training process. Here, we provide Theorem 2 about the relationship between the teacher's temperature and the student's class-wise error risk gap.

**Theorem 2.** *If the teacher $f_t^{'}$ has a correct prediction distribution, the teacher temperature $\tau_{c+}^t$ of hard class $c+$ is positively correlated with the error risk gap for student $f_s$, and the teacher temperature $\tau_{c-}^t$ of easy class $c-$ is negatively correlated with the error risk gap for student $f_s$.*

The proof of Theorem 2 can be found in Appendix A.2. In particular, just as the conclusion in [38]: The teacher has a more correct prediction distribution than the student even in the worst classes, which means Theorem 2 holds in most cases. Theorem 2 demonstrates that the different temperatures can influence the student robust fairness: when the student's error risk of $k$-th class is larger than the average error risk, we think that this type of class is relatively hard compared with others, then the teacher temperature for $k$-th class will reduce and the smoothness degree of soft labels will be sharper, and the optimization gap between teacher distribution and student distribution in $k$-th class will corresponding increase, leading to stronger learning intensity for $k$-th class and final reduce the student's class-wise optimization error risk gap.

To achieve the above optimization goal, we adjust the teacher's $k$-th class temperature $\tilde{\tau}_k^t$ for the guidance of adversarial examples as follows:

$$\tilde{\tau}_k^t = \tilde{\tau}_k^t - \beta \cdot \frac{R(f_s(\tilde{x}_k)) - \frac{1}{C}\sum_{i=1}^{C} R(f_s(\tilde{x}_i))}{max(|R(f_s(\tilde{x}_k)) - \frac{1}{C}\sum_{i=1}^{C} R(f_s(\tilde{x}_i))|)}, \tag{10}$$

where $\beta$ is the learning rate, $max(.)$ denotes taking the maximum value, and $|.|$ represents taking the absolute value, $max(|.|)$ is applied for regularization to maintain the stability of optimization. The update operation in Eq.(10) can change the teacher temperature $\tilde{\tau}_k^t$ based on the gap between the student's $k$-th class error risk $R(f_s(\tilde{x}_k))$ and the average error risk $\frac{1}{C}\sum_{i=1}^{C} R(f_s(\tilde{x}_i))$.

Meanwhile, according to [39], both clean and adversarial examples exist the fairness problems and can affect each other, so it is necessary to achieve fairness for both types of data. Since clean and

---

**Algorithm 1** Overview of ABSLD

---

**Require:** the train dataset $\mathcal{D}$, Student $f_s$ with random initial weight $\theta_s$ and temperature $\tau_s$, pretrained robust teacher $f_t$, the initial temperature $\tau_y^t$ and $\tilde{\tau}_y^t$ for the teacher's soft labels for clean examples $x$ and adversarial examples $\tilde{x}$, where $y = \{1, \ldots, C\}$, the max training epochs $max\text{-}epoch$.

1: **for** 0 to $max\text{-}epoch$ **do**
2:     **for** $k \ in \ y = \{1, \ldots, C\}$ **do**
3:         $R(f_s(\tilde{x}_k)) = 0, \ R(f_s(x_k)) = 0.$   *// Initialize the clean and robust error risk for each class.*
4:     **end for**
5:     **for** $Every \ minibatch(x, y) \ in \ \mathcal{D}$ **do**
6:         $\tilde{x} = \underset{||\tilde{x}-x||\leq\epsilon}{argmax} \, KL(f_s(\tilde{x}; \tau^s), f_t'(x; \tilde{\tau}_y^t)).$   *// Get adversarial examples with teacher's soft labels.*
7:         $\theta_s = \theta_s - \eta \cdot \nabla_\theta \mathcal{L}_{absld}(\tilde{x}, x; f_s, f_t').$   *// Update student weight $\theta_s$ with teacher's soft labels.*
8:         $R(f_s(\tilde{x}_y)) = R(f_s(\tilde{x}_y)) + CE(f_s(\tilde{x}), y).$   *// Calculate robust error risk for each class.*
9:         $R(f_s(x_y)) = R(f_s(x_y)) + CE(f_s(x), y).$   *// Calculate clean error risk for each class.*
10:     **end for**
11:     **for** $k \ in \ y = \{1, \ldots, C\}$ **do**
12:         Update $\tilde{\tau}_k^t$ and $\tau_k^t$ based on Eq.(10).   *// Re-temperate teacher's soft labels for $\tilde{x}$ and $x$.*
13:     **end for**
14: **end for**

---

adversarial examples of the same classes may have different error risks during the training process, it is unreasonable to use the same set of class temperatures to adjust both clean and adversarial examples. Here we simultaneously optimize the student's clean error risk $R(f_s(x_k))$ and the student's robust error risk $R(f_s(\tilde{x}_k))$, in other words, we apply two different sets of teacher temperatures: $\tau_k^t$ and $\tilde{\tau}_k^t$, for the adjustment of the teacher's soft labels for clean and adversarial examples, respectively.

Then we extend the Anti-Bias Soft Label Distillation based on [45] and the loss function $\mathcal{L}_{absld}$ in Eq.(8) can be formulated as follows:

$$\mathcal{L}_{absld}(\tilde{x}, x; f_s, f_t') = \alpha \frac{1}{C} \sum_{i=1}^{C} KL(f_s(\tilde{x}_i; \tau^s), f_t'(x_i; \tilde{\tau}_i^t)) + (1-\alpha) \frac{1}{C} \sum_{i=1}^{C} KL(f_s(x_i; \tau^s), f_t'(x_i; \tau_i^t)),$$

$$(11)$$

where $KL$ represents Kullback–Leibler divergence loss, $\alpha$ is the trade-off parameter between accuracy and robustness, $f(x; \tau)$ denotes model $f$ predicts the output probability of $x$ with temperature $\tau$ in the final softmax layer. It should be mentioned that the teacher is frozen and we apply the teacher's predicted soft labels $f_t'(x_k; \tilde{\tau}_k^t)$ for $k$-th class to generate adversarial examples $\tilde{x}_k$ as follows:

$$\tilde{x}_k = \underset{||\tilde{x}_k-x_k||\leq\epsilon}{argmax} \, KL(f_s(\tilde{x}_k; \tau^s), f_t'(x_k; \tilde{\tau}_k^t)),$$

$$(12)$$

and the complete process can be viewed in Algorithm 1.

## 5 Experiments

### 5.1 Experimental Settings

We conduct our experiments on three datasets: CIFAR-10 [16], CIFAR-100, and Tiny-ImageNet [17]. The results about CIFAR-100 and Tiny-ImageNet are in Appendix A.4 and A.5, respectively.

**Baselines.** We consider the standard training method and eight state-of-the-art methods as comparison methods: **AT methods**: SAT [20], and TRADES [41]; **ARD methods**: RSLAD [45], and AdaAD [12]; **Robust Fairness methods**: FRL [39], BAT [28], CFA [36], and Fair-ARD [38].

**Student and Teacher Networks.** For the student model, here we consider two networks for CIFAR-10 and CIFAR-100 including ResNet-18 [10] and MobileNet-v2 [26]. For the teacher model, we follow the setting in [45], and we select WiderResNet-34-10 [40] trained by [41] for CIFAR-10 and WiderResNet-70-16 trained by [9] for CIFAR-100. The teacher's performance is in Appendix A.7.

**Training Setting.** For ABSLD, we train the model using the Stochastic Gradient Descent (SGD) optimizer with an initial learning rate of 0.1, a momentum of 0.9, and a weight decay of 2e-4. The

Table 1: Result in average robustness(%) (Avg.↑), worst robustness(%) (Worst↑), and normalized standard deviation (NSD↓) on CIFAR-10 of ResNet-18.

| Method | Clean | | | FGSM | | | PGD | | | $CW_\infty$ | | | AA | | |
|---|---|---|---|---|---|---|---|---|---|---|---|---|---|---|---|
| | Avg. | Worst | NSD | Avg. | Worst | NSD | Avg. | Worst | NSD | Avg. | Worst | NSD | Avg. | Worst | NSD |
| Natural | 94.57 | 86.30 | 0.035 | 18.60 | 9.00 | 0.436 | 0 | 0 | - | 0 | 0 | - | 0 | 0 | - |
| SAT[20] | 84.03 | 63.90 | 0.118 | 56.71 | 26.70 | 0.283 | 49.34 | 21.00 | 0.332 | 48.99 | 19.90 | 0.352 | 46.44 | 16.80 | 0.385 |
| TRADES[41] | 81.45 | 67.60 | 0.113 | 56.65 | 36.60 | 0.267 | 51.78 | 30.40 | 0.301 | 49.15 | 27.10 | 0.341 | 48.17 | 25.90 | 0.350 |
| RSLAD[45] | 82.94 | 66.30 | 0.122 | 59.51 | 34.70 | 0.244 | 54.00 | 28.50 | 0.276 | **52.51** | 27.00 | 0.296 | **51.25** | 25.50 | 0.304 |
| AdaAD[12] | 84.73 | 68.10 | 0.114 | 59.70 | 34.80 | 0.246 | 53.82 | 29.30 | 0.285 | 52.30 | 26.00 | 0.312 | 50.91 | 24.70 | 0.322 |
| FRL[39] | 82.29 | 64.60 | 0.114 | 55.03 | 37.10 | 0.230 | 49.05 | 31.70 | 0.248 | 47.88 | 30.40 | 0.266 | 46.54 | 28.10 | 0.280 |
| BAT[28] | 86.72 | 72.30 | 0.092 | **60.97** | 33.80 | 0.255 | 49.60 | 22.70 | 0.325 | 47.49 | 19.50 | 0.354 | 48.18 | 20.70 | 0.341 |
| CFA[36] | 78.64 | 63.60 | 0.123 | 57.95 | 36.80 | 0.231 | 54.42 | 33.30 | 0.258 | 50.91 | 27.50 | 0.288 | 50.37 | 26.70 | 0.296 |
| Fair-ARD[38] | 83.81 | 69.40 | 0.112 | 58.41 | 38.50 | 0.251 | 50.91 | 29.90 | 0.296 | 49.96 | 28.30 | 0.312 | 47.97 | 25.10 | 0.338 |
| **ABSLD** | 83.04 | 68.10 | 0.103 | 59.83 | **40.50** | **0.202** | 54.50 | **36.50** | **0.216** | 51.77 | **32.80** | **0.249** | 50.25 | **31.00** | **0.256** |

Table 2: Result in average robustness(%) (Avg.↑), worst robustness(%) (Worst↑), and normalized standard deviation (NSD↓) on CIFAR-10 of MobileNet-v2.

| Method | Clean | | | FGSM | | | PGD | | | $CW_\infty$ | | | AA | | |
|---|---|---|---|---|---|---|---|---|---|---|---|---|---|---|---|
| | Avg. | Worst | NSD | Avg. | Worst | NSD | Avg. | Worst | NSD | Avg. | Worst | NSD | Avg. | Worst | NSD |
| Natural | 93.35 | 85.20 | 0.036 | 12.21 | 0.90 | 0.750 | 0 | 0 | - | 0 | 0 | - | 0 | 0 | - |
| SAT[20] | 82.30 | 65.80 | 0.131 | 56.19 | 34.60 | 0.279 | 48.52 | 24.90 | 0.334 | 47.22 | 23.80 | 0.361 | 44.38 | 18.50 | 0.410 |
| TRADES[41] | 79.37 | 59.00 | 0.131 | 54.94 | 31.60 | 0.297 | 50.03 | 27.20 | 0.330 | 47.02 | 23.20 | 0.367 | 46.25 | 22.10 | 0.379 |
| RSLAD[45] | 82.96 | 66.30 | 0.130 | **59.84** | 34.30 | 0.256 | **53.88** | 28.20 | 0.288 | **52.28** | 25.70 | 0.316 | 50.67 | 23.90 | 0.333 |
| AdaAD[12] | 83.72 | 66.90 | 0.123 | 57.63 | 33.70 | 0.262 | 51.89 | 26.90 | 0.300 | 50.34 | 24.70 | 0.317 | 48.81 | 22.50 | 0.334 |
| FRL[39] | 81.02 | 70.50 | 0.089 | 53.84 | 40.50 | 0.207 | 47.71 | 35.00 | 0.241 | 44.96 | 30.90 | 0.276 | 43.53 | 28.40 | 0.292 |
| BAT[28] | 83.01 | 70.30 | 0.102 | 53.01 | 32.10 | 0.284 | 44.08 | 25.00 | 0.344 | 41.85 | 21.60 | 0.397 | 42.65 | 23.40 | 0.369 |
| CFA[36] | 80.34 | 64.60 | 0.112 | 56.45 | 32.20 | 0.260 | 52.34 | 28.10 | 0.292 | 48.62 | 23.20 | 0.320 | **50.68** | 23.90 | 0.331 |
| Fair-ARD[38] | 82.44 | 69.40 | 0.100 | 56.29 | 38.60 | 0.226 | 50.91 | 29.90 | 0.263 | 48.18 | 30.80 | 0.286 | 46.62 | 27.60 | 0.302 |
| **ABSLD** | 82.54 | 69.50 | 0.102 | 58.55 | **41.40** | **0.207** | 52.99 | **35.70** | **0.224** | 50.39 | **31.90** | **0.254** | 48.71 | **30.30** | **0.259** |

learning rate $\beta$ of temperature is initially set as 0.1. For CIFAR-10 and CIFAR-100, we set the training epochs to 300. The learning rate is divided by 10 at the 215-th, 260-th, and 285-th epochs; We set the batch size to 128 for both CIFAR-10 and CIFAR-100 following [45]. For the inner maximization, we use a 10-step PGD with a random start size of 0.001 and a step size of 2/255, and the perturbation is bounded to the $L_\infty$ norm $\epsilon = 8/255$. The more training setting can be found in Appendix A.3.

**Metrics.** We apply two metrics to evaluate the robust fairness: Normalized Standard Deviation (NSD[3]) [38] and the worst-class robustness [36]. **NSD can reflect robust fairness while also considering the average robustness**. The smaller standard deviation means better fairness, and the larger average means better robustness, so the smaller NSD means better comprehensive performance in terms of fairness and robustness. **The worst-class robustness** is easy to understand, and a larger value means better fairness. For CIFAR-10, we directly report the worst class robust accuracy; For CIFAR-100 and Tiny-ImageNet, due to the poor performance of the worst class robustness and only 100 images (CIFAR-100) or 50 images (Tiny-ImageNet) for each class in the test set, we follow the operation in CFA [36] and report the worst 10% class robust accuracy. Besides, we also report the **average robustness** as a reference. The attack setting for evaluation can be found in Appendix A.3.

### 5.2 Robust Fairness Performance

The performances of ResNet-18 and MobileNet-v2 trained by our ABSLD and other baseline methods under the various attacks are shown in Table 1, Table 2 for CIFAR-10. The results demonstrate that ABSLD achieves the state-of-the-art worst-class robustness on CIFAR-10. For ResNet-18 on CIFAR-10, ABSLD improves the worst class robustness by 2.0%, 3.2%, 2.4%, and 2.9% compared with the best baseline method against the FGSM, PGD, $CW_\infty$, and AA. Moreover, ABSLD shows relevant superiority on MobileNet-v2 compared with other methods.

Moreover, ABSLD can also show the best comprehensive performance of fairness and robustness (NSD) on CIFAR-10. For ResNet-18 on CIFAR-10, ABSLD reduces the NSD by 0.028, 0.032, 0.017, and 0.024 compared with the best baseline method against the FGSM, PGD, $CW_\infty$, and AA. The result indicates that although the trade-off between robustness and fairness still exists as [19] say, we obtain the highest robust fairness while sacrificing the least average robustness.

Meanwhile, we visualize the class-wise robustness in Figure 3, and the result shows that the robustness of harder classes (class 3, 4, 5, 6) have different levels of improvement, which demonstrates that our method is beneficial to the overall robust fairness but not only to the worst class. Moreover, combined

---

[3]NSD is a metric applied in [38] to measure the robust fairness. NSD = SD/Avg., where SD is the Standard Deviation of class-wise robustness and Avg. is the average robustness.

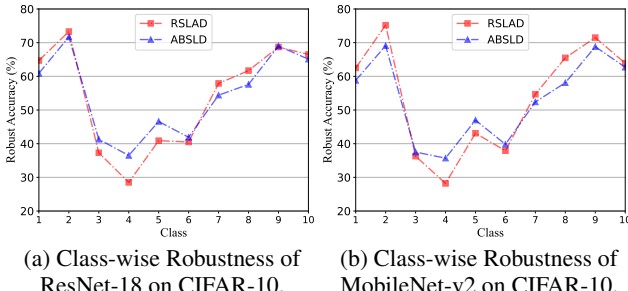

(a) Class-wise Robustness of ResNet-18 on CIFAR-10.

(b) Class-wise Robustness of MobileNet-v2 on CIFAR-10.

Figure 3: The class-wise robustness (PGD) of models guided by RSLAD and ABSLD on CIFAR-10. We can see that the harder classes' robustness (class 3, 4, 5, 6) of ABSLD (blue lines) have different levels of improvement compared with RSLAD (red lines).

with Figure 2, we can find that the trend of class-wise bias is similar in different training strategies, indicating that the bias may be sourced from the dataset itself, which further confirms Corollary 1.

In particular, we compare ABSLD with Fair-ARD [38], which is an adaptive re-weighting method on ARD. From the results, ABSLD has better robust fairness performance, which means that our proposed re-temperating method has superiority compared to the re-weighting method.

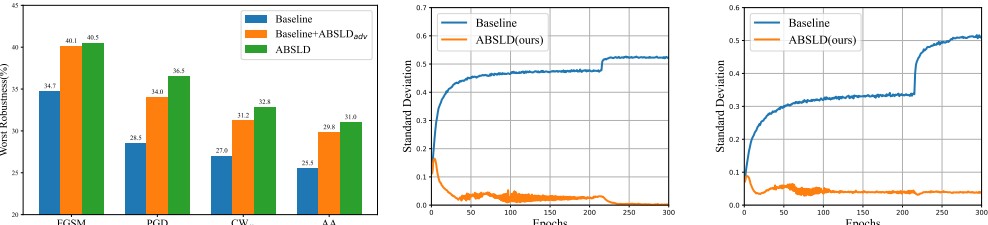

Figure 4: Ablation study for Baseline, Baseline+ABSLD$_{adv}$, and ABSLD.

Figure 5: Standard deviation of class-wise clean optimization error risk.

Figure 6: Standard deviation of class-wise adversarial optimization error risk.

## 5.3 Ablation Study

To certify the effectiveness of our method, we perform ablation experiments on every component of ABSLD. First, based on the baseline method [45], we re-temperate the teacher's soft labels for the adversarial examples but do not re-temperate the teacher's soft labels for the clean examples (Baseline+ABSLD$_{adv}$); then we re-temperate the teacher's soft labels for both the adversarial examples and clean examples (ABSLD). The results are shown in Figure 4. The results demonstrate the effectiveness of our ABSLD, and pursuing fairness for clean examples can also help robust fairness for adversarial examples as claimed in [39].

Meanwhile, to verify the optimization effect of our method, we visualize the standard deviation of class-wise optimization error risk in the training process (the optimization error risk is normalized by dividing the mean), which can reflect the optimization gap between different classes. We visualize the standard deviation of both the clean and adversarial optimization error risk, and the results are shown in Figure 5 and Figure 6. We can notice that the standard deviation of the baseline increases as the training epoch increases, which demonstrates that the baseline pays more attention to reducing the error risk of easy class, but the error risk of hard class is ignored, eventually leading to robust unfairness. On the contrary, our ABSLD can remarkably reduce the standard deviation of student's class-wise optimization error risk, which demonstrates the effectiveness of our method.

# 6  Conclusion

In this paper, we comprehensively explored the potential factors that influence robust fairness in the model optimization process. We first found that the smoothness degrees of soft labels for different classes can be applied to eliminate the robust fairness based on empirical observation and theoretical analysis. Then we proposed Anti-Bias Soft Label Distillation (ABSLD) to address the robust fairness problem by adjusting the class-wise smoothness degree of soft labels. We adjusted the teacher's soft labels by assigning different temperatures to different classes based on the performance of student's class-wise error risk. A series of experiments proved that ABSLD was superior to state-of-the-art methods in the comprehensive metric (NSD) of robustness and fairness.

## Acknowledgement

This work was supported by Alibaba Group through Alibaba Reasearch Intern Program, the Project of the National Natural Science Foundation of China (No.62076018), and the Fundamental Research Funds for the Central Universities.

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

# A Appendix

## A.1 The proof of Theorems 1 in Sec. 3

For the initial state of the DNN model $f$ with random distribution $I$, the DNN model has no preference for any examples. So for the dataset containing hard class $c_+$ and easy class $c_-$, we have a relationship between class-wise error risk as follows:

$$R(f(x_{c_+}; \theta_I)) = R(f(x_{c_-}; \theta_I)), \tag{13}$$

in other words, the model has the same error risk for easy class $c_-$ and hard class $c_+$, so we set the model's prediction distribution $f(x; \theta_I) = \{p_{c_-}^I(x), p_{c_+}^I(x)\}$ and has the relationship as follows:

$$\mathbb{E}(p^I(x)) = \mathbb{E}(p_{c_+}^I(x_{c_+})) = \mathbb{E}(p_{c_-}^I(x_{c_-})) = 0.5. \tag{14}$$

Then we analyze the process of knowledge distillation, we assume that DNN model $f$ is optimized with the guidance of the soft label distribution with the same smoothness degree $P_{\lambda 1}$, which satisfies:

$$p_{c_-}^{\lambda 1}(x_{c_-}) = p_{c_+}^{\lambda 1}(x_{c_+}) > 0.5 > p_{c_-}^{\lambda 1}(x_{c_+}) = p_{c_+}^{\lambda 1}(x_{c_-}). \tag{15}$$

And the updated parameter $\theta_{\lambda 1}$ guided by the soft label distribution $P_{\lambda 1}$ can formulated as follows:

$$\theta_{\lambda 1} = \theta_I - \eta \cdot \frac{\partial KL(f(x; \theta_I), P_{\lambda 1})}{\partial \theta}, \tag{16}$$

here we divide the partial derivative of the optimization function $KL$ with respect to the student parameter $\theta$ into two parts:

$$\frac{\partial KL(f(x; \theta_I), P_{\lambda 1})}{\partial \theta} = \sum_{c=1}^{C=2} \frac{\partial z_c(x)}{\partial \theta} \cdot \frac{\partial KL(f(x; \theta_I), P_{\lambda 1})}{\partial z_c(x)}, \tag{17}$$

where $z_c(x)$ denotes the $c$-th dimension output logits of the model before the softmax layer (denote the prediction logits of $c$-th class).

The first part $\frac{\partial z_c(x)}{\partial \theta}$ can be regarded as the impact of the class itself on model weight optimization, in a practical sense, this part reflects the inner relationship between DNN and sample, more specifically, it can reflect the difficulty of the sample itself for the model and not relate to the optimization object.

The second part $\frac{\partial KL(f(x; \theta_I), P_{\lambda 1})}{\partial z_c(x)}$ can be regarded as the impact of the optimization object on the different classes. Although adjusting the optimization goal is not enough to change the bias derived from the sample itself, it can be adjusted to make the model perform as fair as possible.

Here we further extend the partial derivative of optimization object $KL(f(x; \theta_I), P_{\lambda 1})$ toward the prediction logit $z_c(x)$ of class $c$ and obtain:

$$\frac{\partial KL(f(x; \theta_I), P_{\lambda 1})}{\partial z_c(x)}$$
$$= \sum_{i=1}^{C=2} \frac{\partial p_i(x)}{\partial z_c(x)} \cdot \frac{\partial KL(f(x; \theta_I), P_{\lambda 1})}{\partial p_i(x)}$$
$$= \sum_{i=1}^{C=2} p_i^{\lambda 1}(x) p_c^I(x) - p_c^{\lambda 1}(x)$$
$$= p_c^I(x) - p_c^{\lambda 1}(x), \tag{18}$$

so we can easily obtain:

$$\mathbb{E}\left(\frac{\partial KL(f(x_{c-}; \theta_I), P_{\lambda 1})}{\partial z_{c-}(x_{c-})}\right) = \mathbb{E}\left(\frac{\partial KL(f(x_{c+}; \theta_I), P_{\lambda 1})}{\partial z_{c+}(x_{c+})}\right)$$
$$= \mathbb{E}(p_c^I(x_c) - p_{c-}^{\lambda 1}(x_{c-})) = \mathbb{E}(p_c^I(x_c) - p_{c+}^{\lambda 1}(x_{c+})), \tag{19}$$

$$\mathbb{E}(\frac{\partial KL(f(x_{c+};\theta_I),P_{\lambda 1})}{\partial z_{c-}(x_{c+})}) = \mathbb{E}(\frac{\partial KL(f(x_{c-};\theta_I),P_{\lambda 1})}{\partial z_{c+}(x_{c-})})$$
$$= \mathbb{E}(p_c^I(x_c) - p_{c-}^{\lambda 1}(x_{c+})) = \mathbb{E}(p_c^I(x_c) - p_{c+}^{\lambda 1}(x_{c-})). \tag{20}$$

Based on the Corollary 3.1, the DNN model with parameter $\theta_{\lambda 1}$ has different the error risk of the hard class $c+$ and the easy class $c-$, so we have:

$$R(f(x_{c_+};\theta_{\lambda 1})) > R(f(x_{c_-};\theta_{\lambda 1})), \tag{21}$$

from the optimization perspective, the optimization gradient toward the model parameter for different classes directly influences the class error risk, more specifically, the easy class error risk is less than the hard class error risk, so the gradient expectation of the easy class is higher than the gradient expectation of hard class, and the proof is as follows:

In the initial state, we assume that the model is a uniform distribution, and in this case, the error optimization risk is the same for the easy and hard classes:

$$\mathbb{E}(KL(f(x_{c-};\theta_I),P_{\lambda 1})) = \mathbb{E}(KL(f(x_{c+};\theta_I),P_{\lambda 1})), \tag{22}$$

then we will simplify the optimization of the model into a one-step gradient iteration process, which means we only consider the initial state before optimization and the last state after optimization. The model parameter is updated from initial $\theta_I$ to optimized $\theta_{opt}$, since easy classes perform better than hard classes after the optimization process, the easy class error risk is smaller than hard class error risk:

$$\mathbb{E}(KL(f(x_{c-};\theta_{opt}),P_{\lambda 1})) < \mathbb{E}(KL(f(x_{c+};\theta_{opt}),P_{\lambda 1}))$$
$$< \mathbb{E}(KL(f(x_{c-};\theta_I),P_{\lambda 1})) = \mathbb{E}(KL(f(x_{c+};\theta_I),P_{\lambda 1})), \tag{23}$$

based on the above result, we can have the result as follows:

$$\mathbb{E}(|KL(f(x_{c-};\theta_{opt}),P_{\lambda 1}) - KL(f(x_{c-};\theta_I),P_{\lambda 1})|)$$
$$> \mathbb{E}(|KL(f(x_{c+};\theta_{opt}),P_{\lambda 1}) - KL(f(x_{c+};\theta_I),P_{\lambda 1})|) \tag{24}$$

at this time, we assume that model's parameter $\theta$ is differentiable and continuous, then we can approximate the gradient expectation of the partial derivative $\mathbb{E}(\frac{\partial KL(f(x_{c-};\theta_I),P_{\lambda 1})}{\partial \theta})$ and $\mathbb{E}(\frac{\partial KL(f(x_{c+};\theta_I),P_{\lambda 1})}{\partial \theta})$ as follows:

$$\mathbb{E}(\frac{\partial KL(f(x_{c-};\theta_I),P_{\lambda 1})}{\partial \theta})$$
$$\approx \mathbb{E}(\frac{KL(f(x_{c-};\theta_{opt}),P_{\lambda 1}) - KL(f(x_{c-};\theta_I),P_{\lambda 1})}{\theta_{opt} - \theta_I}), \tag{25}$$

$$\mathbb{E}(\frac{\partial KL(f(x_{c+};\theta_I),P_{\lambda 1})}{\partial \theta})$$
$$\approx \mathbb{E}(\frac{KL(f(x_{c+};\theta_{opt}),P_{\lambda 1}) - KL(f(x_{c+};\theta_I),P_{\lambda 1})}{\theta_{opt} - \theta_I}), \tag{26}$$

combined with the relationship between easy class error risk and hard class error risk, we can obtain the result that the gradient absolute value expectation of partial derivative about the easy class's optimization goal toward the model parameter ($\mathbb{E}(|\frac{\partial KL(f(x_{c-};\theta_I),P_{\lambda 1})}{\partial \theta}|)$) is higher than the gradient absolute value expectation of partial derivative about the hard class's optimization goal toward the model parameter ($\mathbb{E}(|\frac{\partial KL(f(x_{c+};\theta_I),P_{\lambda 1})}{\partial \theta}|)$) as follows:

$$\mathbb{E}(|\frac{\partial KL(f(x_{c-};\theta_I),P_{\lambda 1})}{\partial \theta}|) > \mathbb{E}(|\frac{\partial KL(f(x_{c+};\theta_I),P_{\lambda 1})}{\partial \theta}|), \tag{27}$$

so we can obtain the assumption: If the model is a uniform distribution before the optimization process and the easy class error risk is less than the hard class error risk after optimization process,

then the gradient expectation absolute value of partial derivative about the easy class's optimization goal toward the model parameter is higher than the gradient expectation absolute value of partial derivative about the hard class's optimization goal toward the model parameter.

Then we can decouple the gradient into two parts based on the class types: $\frac{\partial KL(f(x_{c-};\theta_I),P_{\lambda 1})}{\partial \theta}$ and $\frac{\partial KL(f(x_{c+};\theta_I),P_{\lambda 1})}{\partial \theta}$.

Here we take the hard class as an example, the gradient can be divided into two parts: $\frac{\partial z_{c+}(x_{c+})}{\partial \theta}\frac{\partial KL(f_s(x_{c+};\theta_I),P_{\lambda 1})}{\partial z_{c+}}$ and $\frac{\partial z_{c-}(x_{c+})}{\partial \theta}\frac{\partial KL(f_s(x_{c+};\theta_I),P_{\lambda 1})}{\partial z_{c-}}$, where the first part represents the ability to make hard class's samples more like hard class, and the second part represents the ability to make hard class's samples less like easy class. The final prediction performance of hard class $c+$ is influenced by both parts of the optimization gradient due to the softmax operation.

Based on the above analysis, the relationship between the sum value of the above two parts of gradient for hard class ($\nabla^+_{f\sim\lambda 1}$) and easy class ($\nabla^-_{f\sim\lambda 1}$) can be assumed under a more stringent condition as follows:

$$
\begin{aligned}
\nabla^-_{f\sim\lambda 1} = \mathbb{E}(|\frac{\partial z_{c-}(x_{c-})}{\partial \theta}\frac{\partial KL(f_s(x_{c-};\theta_I),P_{\lambda 1})}{\partial z_{c-}}| + |\frac{\partial z_{c+}(x_{c-})}{\partial \theta}\frac{\partial KL(f_s(x_{c-};\theta_I),P_{\lambda 1})}{\partial z_{c+}}|) > \\
\nabla^+_{f\sim\lambda 1} = \mathbb{E}(|\frac{\partial z_{c-}(x_{c+})}{\partial \theta}\frac{\partial KL(f_s(x_{c+};\theta_I),P_{\lambda 1})}{\partial z_{c-}}| + |\frac{\partial z_{c+}(x_{c+})}{\partial \theta}\frac{\partial KL(f_s(x_{c+};\theta_I),P_{\lambda 1})}{\partial z_{c+}}|).
\end{aligned}
\tag{28}
$$

Combined with Eq.(19), Eq.(20) and Eq.(28), we have a relationship about the derivative of $z_c$ toward model parameter $\theta$, which can reflect the bias of sample for the model:

$$
\mathbb{E}(|\frac{\partial z_{c-}(x_{c-})}{\partial \theta}| + |\frac{\partial z_{c+}(x_{c-})}{\partial \theta}|) > \mathbb{E}(|\frac{\partial z_{c-}(x_{c+})}{\partial \theta}| + |\frac{\partial z_{c+}(x_{c+})}{\partial \theta}|).
\tag{29}
$$

After the analysis of the model characteristic with the guidance of soft label distribution $P_{\lambda 1}$, we try to compare the model characteristic difference between soft label distribution $P_{\lambda 1}$ and soft label distribution $P_{\lambda 2}$. Initially, we can obtain the following relationship for the probabilities of different distributions:

$$
\begin{aligned}
p^{\lambda 2}_{c+}(x_{c+}) = p^{\lambda 1}_{c+}(x_{c+}) + v_1 > p^{\lambda 1}_{c+}(x_{c+}) = \\
p^{\lambda 1}_{c-}(x_{c-}) > p^{\lambda 2}_{c-}(x_{c-}) = p^{\lambda 1}_{c-}(x_{c-}) - v_2 > 0.5,
\end{aligned}
\tag{30}
$$

then we further analyze the model class-wise error risk gap guided by label distribution. The total optimization gradient for different classes can reflect the bias degree, so we get the class optimization gradient gap guided by label distribution $P_{\lambda 1}$ and label distribution $P_{\lambda 2}$, respectively:

$$
\begin{aligned}
\mathcal{B}_{f\sim\lambda 1} &= \nabla^-_{f\sim\lambda 1} - \nabla^+_{f\sim\lambda 1} \\
&= \mathbb{E}(|\frac{\partial z_{c-}(x_{c-})}{\partial \theta}\frac{\partial KL(f_s(x_{c-};\theta_I),P_{\lambda 1})}{\partial z_{c-}}| + |\frac{\partial z_{c+}(x_{c-})}{\partial \theta}\frac{\partial KL(f_s(x_{c-};\theta_I),P_{\lambda 1})}{\partial z_{c+}}| - \\
&\quad |\frac{\partial z_{c-}(x_{c+})}{\partial \theta}\frac{\partial KL(f_s(x_{c+};\theta_I),P_{\lambda 1})}{\partial z_{c-}}| - |\frac{\partial z_{c+}(x_{c+})}{\partial \theta}\frac{\partial KL(f_s(x_{c+};\theta_I),P_{\lambda 1})}{\partial z_{c+}}|) \\
&= \mathbb{E}(|\frac{\partial z_{c-}(x_{c-})}{\partial \theta}|(p^{\lambda 1}_{c-}(x_{c-}) - p^I_c) + |\frac{\partial z_{c+}(x_{c-})}{\partial \theta}|(p^I_c - p^{\lambda 1}_{c+}(x_{c-})) - \\
&\quad |\frac{\partial z_{c-}(x_{c+})}{\partial \theta}|(p^I_c - p^{\lambda 1}_{c-}(x_{c+})) - |\frac{\partial z_{c+}(x_{c+})}{\partial \theta}|(p^{\lambda 1}_{c+}(x_{c+}) - p^I_c)),
\end{aligned}
\tag{31}
$$

$$
\begin{aligned}
\mathcal{B}_{f\sim\lambda 2} &= \nabla^-_{f\sim\lambda 2} - \nabla^+_{f\sim\lambda 2} \\
&= \mathbb{E}(|\frac{\partial z_{c-}(x_{c-})}{\partial \theta}|(p^{\lambda 2}_{c-}(x_{c-}) - p^I_c) + |\frac{\partial z_{c+}(x_{c-})}{\partial \theta}|(p^I_c - p^{\lambda 2}_{c+}(x_{c-})) - \\
&\quad |\frac{\partial z_{c-}(x_{c+})}{\partial \theta}|(p^I_c - p^{\lambda 2}_{c-}(x_{c+})) - |\frac{\partial z_{c+}(x_{c+})}{\partial \theta}|(p^{\lambda 2}_{c+}(x_{c+}) - p^I_c)),
\end{aligned}
\tag{32}
$$

if the model parameter $\theta_{\lambda 2}$ still satisfies: $R(f(x_{c_+};\theta_{\lambda 2})) > R(f(x_{c_-};\theta_{\lambda 2}))$, then we can obtain $\mathcal{B}_{f\sim\lambda 2} > 0$, and the relationship between $\mathcal{B}_{f\sim\lambda 1}$ and $\mathcal{B}_{f\sim\lambda 2}$ is as follows:

$$
\begin{aligned}
&\mathcal{B}_{f\sim\lambda 1} - \mathcal{B}_{f\sim\lambda 2} \\
=&\mathbb{E}(|\frac{\partial z_{c-}(x_{c-})}{\partial\theta}|(p_{c-}^{\lambda 1}(x_{c-}) - p_{c-}^{\lambda 2}(x_{c-})) + |\frac{\partial z_{c+}(x_{c-})}{\partial\theta}|(p_{c+}^{\lambda 2}(x_{c-}) - p_{c+}^{\lambda 1}(x_{c-})) \\
&- |\frac{\partial z_{c-}(x_{c+})}{\partial\theta}|(-p_{c-}^{\lambda 2}(x_{c+}) + p_{c-}^{\lambda 1}(x_{c+})) - |\frac{\partial z_{c+}(x_{c+})}{\partial\theta}|(p_{c+}^{\lambda 2}(x_{c+}) - p_{c+}^{\lambda 1}(x_{c+}))) \\
=&\mathbb{E}(|\frac{\partial z_{c-}(x_{c-})}{\partial\theta}|v_2 + |\frac{\partial z_{c+}(x_{c-})}{\partial\theta}|v_2 + |\frac{\partial z_{c-}(x_{c+})}{\partial\theta}|v_1 + |\frac{\partial z_{c+}(x_{c+})}{\partial\theta}|v_1) > 0,
\end{aligned}
\tag{33}
$$

so we can have the following conclusion:

$$
\mathcal{B}_{f\sim\lambda 1} > \mathcal{B}_{f\sim\lambda 2}.
\tag{34}
$$

Based on the above results, the model total gradient expectation gap of different classes trained by soft label distribution $P_{\lambda 1}$ is larger than gradient expectation gap between different classes trained by soft label distribution $P_{\lambda 2}$, so the bias degree of model trained by the soft label distribution $P_{\lambda 1}$ is greater than by the bias degree of model trained by the guidance of soft label distribution $P_{\lambda 2}$, we can get the conclusion as follows:

$$
R(f(x_{c_+};\theta_{\lambda 1})) - R(f(x_{c_-};\theta_{\lambda 1})) > R(f(x_{c_+};\theta_{\lambda 2})) - R(f(x_{c_-};\theta_{\lambda 2})).
\tag{35}
$$

Then the Theorem 1 is proved.

## A.2 The proof of Theorems 2 in Sec. 4

Based on the correct prediction distribution assumption about the teacher model, we can obtain the relationship of probability toward $k$-th class as follows:

$$
\mathbb{E}(p_k^t(x_k;\tau_k^t)) > \mathbb{E}(p_{c\neq k}^t(x_k;\tau_k^t)).
\tag{36}
$$

Then we extend the temperature into the mathematical formula of the prediction probability, we can obtain:

$$
p_k^t(x_k;\tau_k^t) = \frac{exp(z_k(x_k)/\tau_k^t)}{\sum_{j=1}^{C} exp(z_j(x_k)/\tau_k^t)},
\tag{37}
$$

where $z_k(x)$ denotes the $k$-th dimension output logits of model before softmax layer. Since $\sum_{j=1}^{C} exp(z_j(x_k)/\tau_k^t)$ is applied as a normalization application, here we mainly focus the change $exp(z(x_k)/\tau_k^t)$ with the temperature $\tau_k^t$. $exp(\cdot)$ is a monotonic increasing function, so based on the Eq.(36), we can easily obtain:

$$
\mathbb{E}(z_k(x_k)) > \mathbb{E}(z_{c\neq k}(x_k)),
\tag{38}
$$

here we mainly focus the change $exp(z(x_k)/\tau_k^t)$ with the temperature $\tau_k^t$. We assume the $\tau_k^t$ increase into $\tau_k^t + \Delta_k^\tau (\Delta_k^\tau > 0)$, then we have the partial derivative of $exp(z(x_k)/\tau_k^t) - exp(z(x_k)/(\tau_k^t + \Delta_k^\tau))$ with respect to $z(x_k)$ as follows:

$$
\begin{aligned}
&\frac{\partial(exp(z(x_k)/\tau_k^t) - exp(z(x_k)/(\tau_k^t + \Delta_k^\tau)))}{\partial(z(x_k))}, \\
=&\frac{(exp(z(x_k)/\tau_k^t)}{\tau_k^t} - \frac{(exp(z(x_k)/(\tau_k^t + \Delta_k^\tau)))}{\tau_k^t + \Delta_k^\tau} > 0,
\end{aligned}
\tag{39}
$$

so we have the conclusion as follows:

$$
\begin{aligned}
exp(z_k(x_k)/\tau_k^t) - exp(z_k(x_k)/(\tau_k^t + \Delta_k^\tau)) > \\
exp(z_{c\neq k}(x_k)/\tau_k^t) - exp(z_{c\neq k}(x_k)/(\tau_k^t + \Delta_k^\tau)),
\end{aligned}
\tag{40}
$$

then we can have relationship between the prediction distribution of different temperature as follows:

$$
\begin{aligned}
\mathbb{E}(p_k^t(x_k;\tau_k^t)) - \mathbb{E}(p_{c\neq k}^t(x_k;\tau_k^t)) > \\
\mathbb{E}(p_k^t(x_k;\tau_k^t + \Delta_k^\tau)) - \mathbb{E}(p_{c\neq k}^t(x_k;\tau_k^t + \Delta_k^\tau)).
\end{aligned}
\tag{41}
$$

Based on the above analysis, we assume that the teacher's temperature of the easy class and hard class increase into $\tau_{c-}^t + \Delta_{c-}^\tau$ and $\tau_{c+}^t + \Delta_{c+}^\tau$ and obtain teacher model with soft label prediction $f_t'$, then following the Derivation in Theorem 1, we can obtain:

$$
\begin{aligned}
&\mathcal{B}_{f_s \sim f_t} - \mathcal{B}_{f_s \sim f_t'} \\
=&\mathbb{E}(|\frac{\partial z_{c-}(x_{c-})}{\partial \theta}|(p_{c-}^t(x_{c-}; \tau_{c-}^t)) - p_{c-}^t(x_{c-}; \tau_{c-}^t + \Delta_{c-}^\tau)) \\
&+ |\frac{\partial z_{c+}(x_{c-})}{\partial \theta}|(p_{c+}^t(x_{c-}; \tau_{c-}^t + \Delta_{c-}^\tau) - p_{c+}^t(x_{c-}; \tau_{c-}^t)) \\
&- |\frac{\partial z_{c-}(x_{c+})}{\partial \theta}|(p_{c-}^t(x_{c+}; \tau_{c-}^t) - p_{c-}^t(x_{c+}; \tau_{c+}^t + \Delta_{c+}^\tau)) \\
&- |\frac{\partial z_{c+}(x_{c+})}{\partial \theta}|(p_{c+}^t(x_{c+}; \tau_{c+}^t + \Delta_{c+}^\tau) - p_{c+}^t(x_{c+}; \tau_{c-}^t))).
\end{aligned}
\tag{42}
$$

Then we can further analyze the relationship between the temperature and the error risk gap. Here we assume that the teacher's temperature of easy class $\tau_{c-}^k$ is unchanged ($\Delta_{c-}^\tau = 0$), the teacher's temperature of hard class $\tau_{c+}^k$ increases into $\tau_{c+}^k + \Delta_{c+}^\tau$ ($\Delta_{c+}^\tau > 0$), we can obtain:

$$
\mathcal{B}_{f_s \sim f_t} < \mathcal{B}_{f_s \sim f_t'},
\tag{43}
$$

following the analysis in Appendix A.1, DNN model $f_s$ is optimized with the guidance of the teacher's soft label distribution $f_t$ and $f_t'$, and can obtain the model parameter $\theta_t$ and $\theta_t'$, respectively. We can get the conclusion as follows:

$$
R(f_s(x_{c+}; \theta_t)) - R(f_s(x_{c-}; \theta_t) < R(f(x_{c+}; \theta_t')) - R(f(x_{c-}; \theta_t')).
\tag{44}
$$

Then we assume that the teacher's temperature of hard class $\tau_{c+}^k$ is unchanged ($\Delta_{c+}^\tau = 0$), the teacher's temperature of easy class $\tau_{c-}^k$ increases into $\tau_{c-}^k + \Delta_{c-}^\tau$ ($\Delta_{c-}^\tau > 0$), we can obtain:

$$
\mathcal{B}_{f_s \sim f_t} > \mathcal{B}_{f_s \sim f_t'},
\tag{45}
$$

and following the analysis in Appendix A.1, we can get the conclusion as follows:

$$
R(f_s(x_{c+}; \theta_t)) - R(f_s(x_{c-}; \theta_t) > R(f(x_{c+}; \theta_t')) - R(f(x_{c-}; \theta_t')).
\tag{46}
$$

Based on the above results, we can obtain the teacher temperature $\tau_{c+}^t$ of hard class $c+$ is positively correlated with the error risk gap for student $f_s$, and the teacher temperature $\tau_{c-}^t$ of easy class $c-$ is negatively correlated with the error risk gap for student $f_s$.

Then the Theorem 2 is proved.

### A.3 Additional Experimental Setting

As discussed in [36], the worst class robust accuracy changes drastically when the average robustness convergence, so we follow [36] and select the best checkpoint of the highest mean value of all-class average robustness and the worst class robustness (where the worst class for CIFAR-10 and the worst 10% class for CIFAR-100) in all baselines and our method for a fair comparison.

For ABSLD, to maintain stability when adjusting the teacher prediction distribution, we hold the student temperature $\tau^s$ constant, and the student's optimization error risk for each class can be compared under the same standard. The teacher temperature of $\tau_k^t$ and $\tilde{\tau}_k^t$ are initially set as 1 for all the classes. The student temperature $\tau_s$ is set to constant 1 without additional adjustment. With additional instruction, the maximum and minimum values of temperature are 5 and 0.5, respectively; For ResNet-18 on CIFAR-100, the maximum and minimum values of temperature are 3 and 0.8, respectively. All the experiments are conducted in a single GeForce RTX 3090, and our ABSLD takes approximately one GPU day for training a model.

For the baselines, we strictly follow original setting if without additional instruction. For FRL [39], we use the Reweight+Remargin under the threshold of 0.05. For CFA [36], we select the best version (TRADES+CFA) as reported in the original article. For the training of CFA on CIFAR-100, we set

Table 3: Result in average robustness(%) (Avg.↑), worst-10% robustness(%) (Worst↑), and normalized standard deviation (NSD↓) on CIFAR-100 of ResNet-18.

| Method | Clean | | | FGSM | | | PGD | | | $CW_\infty$ | | | AA | | |
|---|---|---|---|---|---|---|---|---|---|---|---|---|---|---|---|
| | Avg. | Worst | NSD | Avg. | Worst | NSD | Avg. | Worst | NSD | Avg. | Worst | NSD | Avg. | Worst | NSD |
| Natural | 75.17 | 53.80 | 0.155 | 7.95 | 0 | 1.106 | 0 | 0 | - | 0 | 0 | - | 0 | 0 | - |
| SAT[20] | 57.18 | 29.30 | 0.292 | 28.95 | 5.60 | 0.607 | 24.56 | 3.20 | 0.682 | 23.78 | 3.10 | 0.697 | 21.78 | 2.30 | 0.747 |
| TRADES[41] | 55.33 | 28.40 | 0.303 | 30.50 | 7.30 | 0.559 | 27.71 | 5.60 | 0.655 | 24.33 | 3.50 | 0.736 | 23.55 | 3.20 | 0.756 |
| RSLAD[45] | 57.88 | 29.40 | 0.302 | 34.50 | 9.20 | 0.550 | 31.19 | 7.40 | 0.598 | 28.13 | 4.70 | 0.669 | 26.46 | 3.70 | 0.703 |
| AdaAD[12] | 58.17 | 29.20 | 0.301 | 34.29 | 8.50 | 0.566 | 30.49 | 6.60 | 0.623 | **28.31** | 5.00 | 0.683 | **26.60** | 4.20 | 0.721 |
| FRL[39] | 55.49 | 30.30 | 0.282 | 28.00 | 7.20 | 0.559 | 24.04 | 5.00 | 0.642 | 22.93 | 3.70 | 0.673 | 21.10 | 2.80 | 0.721 |
| BAT[28] | 62.71 | 34.40 | 0.267 | 33.41 | 7.90 | 0.555 | 28.39 | 5.30 | 0.626 | 23.91 | 3.00 | 0.716 | 22.56 | 2.50 | 0.755 |
| CFA[36] | 59.29 | 33.80 | 0.245 | 33.88 | 10.30 | 0.520 | 31.15 | 8.40 | 0.563 | 26.85 | 5.20 | 0.655 | 25.83 | 4.90 | 0.679 |
| Fair-ARD[38] | 57.19 | 29.40 | 0.303 | 34.06 | 8.40 | 0.558 | 30.57 | 7.00 | 0.598 | 27.96 | 4.50 | 0.666 | 26.15 | 3.90 | 0.713 |
| **ABSLD** | 56.76 | 31.90 | 0.258 | **34.93** | **12.40** | **0.470** | **32.44** | **10.50** | **0.500** | 26.99 | **6.40** | **0.578** | 25.39 | **5.60** | **0.601** |

Table 4: Result in average robustness(%) (Avg.↑), worst-10% robustness(%) (Worst↑), and normalized standard deviation (NSD↓) on CIFAR-100 of MobileNet-v2.

| Method | Clean | | | FGSM | | | PGD | | | $CW_\infty$ | | | AA | | |
|---|---|---|---|---|---|---|---|---|---|---|---|---|---|---|---|
| | Avg. | Worst | NSD | Avg. | Worst | NSD | Avg. | Worst | NSD | Avg. | Worst | NSD | Avg. | Worst | NSD |
| Natural | 74.86 | 53.90 | 0.150 | 5.95 | 0 | 1.423 | 0 | 0 | - | 0 | 0 | - | 0 | 0 | - |
| SAT[20] | 56.70 | 22.90 | 0.338 | 32.10 | 7.00 | 0.580 | 28.61 | 5.30 | 0.650 | 26.55 | 3.50 | 0.706 | 24.36 | 2.40 | 0.764 |
| TRADES[41] | 57.10 | 30.00 | 0.284 | 31.70 | 8.20 | 0.584 | 29.43 | 6.20 | 0.618 | 25.25 | 3.70 | 0.716 | 24.39 | 3.20 | 0.739 |
| RSLAD[45] | 58.71 | 27.10 | 0.311 | **34.30** | 8.30 | 0.565 | 30.58 | 6.30 | 0.621 | **28.11** | 4.30 | 0.683 | **26.32** | 3.30 | 0.724 |
| AdaAD[12] | 54.59 | 24.50 | 0.343 | 31.40 | 6.10 | 0.616 | 27.96 | 4.80 | 0.677 | 25.72 | 2.10 | 0.752 | 23.80 | 1.60 | 0.807 |
| FRL[39] | 56.30 | 26.50 | 0.311 | 31.03 | 8.00 | 0.548 | 27.52 | 6.10 | 0.602 | 25.40 | 3.70 | 0.657 | 23.28 | 2.70 | 0.714 |
| BAT[28] | 65.39 | 38.70 | 0.228 | 33.31 | 5.30 | 0.577 | 27.08 | 2.90 | 0.683 | 22.80 | 1.50 | 0.790 | 21.28 | 1.10 | 0.823 |
| CFA[36] | 59.15 | 34.10 | 0.255 | 32.50 | 8.80 | 0.551 | 29.38 | 7.00 | 0.601 | 25.16 | 4.00 | 0.690 | 23.78 | 3.20 | 0.727 |
| Fair-ARD[38] | 58.97 | 31.40 | 0.289 | 33.76 | 9.30 | 0.542 | 30.07 | 7.30 | 0.607 | 27.64 | 5.60 | 0.651 | 25.79 | 4.00 | 0.700 |
| **ABSLD** | 56.66 | 32.00 | 0.252 | 33.87 | **12.80** | **0.448** | 31.24 | **11.40** | **0.489** | 26.41 | **7.50** | **0.564** | 24.57 | **6.70** | **0.597** |

the fairness threshold to 0.02 for FAWA operation based on the worst-10% class robustness. For Fair-ARD [38], we select the best version (Fair-ARD on CIFAR-10 and Fair-RSLAD on CIFAR-100) as reported in the original article. Due to the different strategy of selecting checkpoint, the baselines may have slight differences from the results in their original papers.

Following previous studies [45; 42; 38], we evaluate the trained model against white-box adversarial attacks: FGSM [8], PGD [20], $CW_\infty$ [3]. For PGD, we apply 20 steps with a step size of 2/255; For $CW_\infty$, we apply 30 steps with a step size of 2/255. Meanwhile, we apply a strong attack: AutoAttack (AA) [5] to evaluate the robustness, which includes four attacks: Auto-PGD (APGD), Difference of Logits Ratio (DLR) attack, FAB-Attack [4], and the black-box Square Attack [1]. The maximum perturbation of all generated adversarial examples is 8/255.

## A.4 The Robustness on CIFAR-100

The performances of ResNet-18 and MobileNet-v2 trained by our ABSLD and other baseline methods under the various attacks are shown in Table 3, Table 4 for CIFAR-100.

The results demonstrate that ABSLD achieves the state-of-the-art worst-class robustness on CIFAR-100. For ResNet-18 on CIFAR-100, ABSLD improves the worst-10% class robustness by 2.1%, 2.1%, 1.2%, and 0.7% compared with the best baseline method against the FGSM, PGD, $CW_\infty$, and AA. Moreover, ABSLD shows relevant superiority on MobileNet-v2 compared with other methods.

Moreover, ABSLD can also show the best comprehensive performance of fairness and robustness (NSD) on CIFAR-100. For ResNet-18 on CIFAR-100, ABSLD reduces the NSD by 0.05, 0.063, 0.077, and 0.078 compared with the best baseline method against the FGSM, PGD, $CW_\infty$, and AA.

## A.5 The Robustness on Tiny-ImageNet

We select the subset of ImageNet: Tiny-ImageNet as the additional dataset. We train with PreActResNet-18 with 100 epochs, while other settings are the same as CIFAR-100. For our ABSLD, the teacher model is PreActResNet-34 trained by TRADES [41], and the maximum and minimnum values of temperature are 1.1 and 0.9. We select RSLAD [45] (the baseline method) and CFA[36] (the second-best method proven in Table 1 and Table 3) as the comparison method. The results in Table 5 show ABSLD has the best performance under the metric of the worst class robustness and NSD under different attacks, verifying our effectiveness and generalization.

Table 5: Result in average robustness(%) (Avg.↑), worst robustness(%) (Worst↑), and normalized standard deviation (NSD↓) on Tiny-ImageNet of PreActResNet-18.

| Method | Clean | | | FGSM | | | PGD | | | CW$_\infty$ | | | AA | | |
|---|---|---|---|---|---|---|---|---|---|---|---|---|---|---|---|
| | Avg. | Worst | NSD | Avg. | Worst | NSD | Avg. | Worst | NSD | Avg. | Worst | NSD | Avg. | Worst | NSD |
| RSLAD | 47.95 | 15.60 | 0.194 | **26.82** | 4.60 | 0.314 | **24.68** | 4.10 | 0.334 | **20.58** | 1.50 | 0.385 | **18.78** | 0.90 | 0.425 |
| CFA | 46.75 | 17.50 | 0.189 | 23.19 | 3.00 | 0.357 | 20.73 | 2.30 | 0.385 | 16.83 | 1.20 | 0.448 | 15.99 | 0.90 | 0.468 |
| **ABSLD** | 47.70 | 17.90 | 0.171 | 25.93 | **5.80** | **0.291** | 23.41 | **4.60** | **0.313** | 19.62 | **3.40** | **0.353** | 17.70 | **2.30** | **0.386** |

## A.6 The Necessity of Adaptive Adjustment

To demonstrate the effectiveness of our self-adaptive temperature adjustment strategy, we manually re-temperate with the static temperature for different classes based on the prior knowledge. Specifically, We set the teacher temperature for difficult class to be small ($\tau_k^t = 0.5$) and set the teacher temperature for easy class to be large ($\tau_k^t = 5$), which is the minimum and maximum values of temperature, respectively. The experiment in Table 6 shows that our adaptive strategy has a better performance than this manual strategy on CIFAR-10. Moreover, the manual strategy needs to be carefully designed and lacks operability for more complex datasets, e.g., 100 classes on CIFAR-100 or 200 classes on Tiny-ImageNet, so we finally apply the adaptive adjustment strategy for ABSLD.

Table 6: Result in average robustness(%) (Avg.↑), worst robustness(%) (Worst↑), and normalized standard deviation (NSD↓) on CIFAR-10 of ResNet-18.

| Method | Clean | | | FGSM | | | PGD | | | CW$_\infty$ | | | AA | | |
|---|---|---|---|---|---|---|---|---|---|---|---|---|---|---|---|
| | Avg. | Worst | NSD | Avg. | Worst | NSD | Avg. | Worst | NSD | Avg. | Worst | NSD | Avg. | Worst | NSD |
| Manual | 81.31 | 66.20 | 0.106 | 57.47 | 34.00 | 0.210 | 49.47 | 27.80 | 0.232 | 48.13 | 25.40 | 0.249 | 45.34 | 24.40 | **0.251** |
| **Adaptive** | 83.04 | 68.10 | 0.103 | **59.83** | **40.50** | **0.202** | **54.50** | **36.50** | **0.216** | **51.77** | **32.80** | **0.249** | **50.25** | **31.00** | 0.256 |

## A.7 The Robustness of Teacher Models

Here we report the performance of Teacher models, we select WiderResNet-34-10 [40] trained by [41] for CIFAR-10 and WiderResNet-70-16 trained by [9] for CIFAR-100 following [45; 43]. For Tiny-ImageNet, the teacher model is PreActResNet-34 trained by TRADES [41]. The performance is shown in Table 7.

Table 7: Robustness (%) of the teachers in our experiments.

| Dataset | Model | Clean | FGSM | PGD | CW$_\infty$ | AA |
|---|---|---|---|---|---|---|
| CIFAR-10 | WideResNet-34-10 | 84.91 | 61.14 | 55.30 | 53.84 | 53.08 |
| CIFAR-100 | WideResNet-70-16 | 60.96 | 35.89 | 33.58 | 31.05 | 30.03 |
| Tiny-ImageNet | PreActResNet-34 | 52.76 | 27.05 | 24.00 | 20.07 | 18.92 |

## A.8 Limitations

At present, although we have improved the fairness of model adversarial robustness with minimal cost, the overall robustness remains unchanged or slightly decreases compared to previous methods in some cases, and how to solve the trade-off between robustness and fairness is one of the directions that will be further explored in the future. Meanwhile, our method is based on the adjustment towards the smoothness degree of soft labels, and cannot be directly applied to adversarial training methods based on one-hot labels but needs to be combined with label smoothing operations.

