# OpenReview forum: "Improving Adversarial Robust Fairness via Anti-Bias Soft Label  Distillation"
_NeurIPS.cc/2024/Conference — NeurIPS 2024 poster_

### Official Review · Reviewer_Dt8R · 2024-07-04

**Soundness:** 2
**Presentation:** 2
**Contribution:** 2
**Rating:** 5
**Confidence:** 5

**Summary:**

This paper aims to improve robust fairness in the adversarial distillation setting.
The proposed method adaptively assigns a smaller temperature for hard classes and a larger temperature for easy classes during the adversarial distillation training.
The smaller temperature means stronger supervision intensity for hard classes, realizing dynamic re-weighting in the training.

Experiments on CIFAR100 and CIFAR-10 show some improvements in the worst-class performance at the cost of overall robustness degradation.

**Strengths:**

(1) The proposed algorithm is easy to implement.
(2) Worst-class performance is enhanced at some cost of overall robustness.

**Weaknesses:**

(1) The proposed algorithm is to allow stronger supervision for hard classes with a smaller temperature during adversarial knowledge distillation.
It seems to have similar effects to re-weighting, i.e., assigning a smaller class weight on hard classes for the KL loss during the knowledge distillation.
What's the difference between the proposed method and such a baseline in nature? Why could the proposed method achieve better results?

(2) Though the proposed method improves the worst class robustness, the overall robustness can't be maintained.

(3) Regarding the proof of theorem 1,
      how do we get the conclusion in Eq. (19) with Eq. (18)?
         From my point of view, the two terms should be the opposites of each other in Eq. (19).

(4) In the proof, the assumption that "the easy class error risk is less than the hard class error risk, so the gradient expectation of the easy class is higher than the gradient expectation of the hard class" is used multiple times.
          However, Why is it established?
          What's the relationship between the gradients and the error risk mathematically?

**Questions:**

(1) The proposed method has close relations with re-weighting.
      The authors should clarify the differences between the proposed method and the re-weighting. Why could the proposed method achieve better performance than re-weighting?

(2) There seem to be some problems in the proof of theorem 1.
     What's the relationship between the gradients and the error risk mathematically?

(3) The proposed method can achieve better worst-class performance. However, the overall robustness degrades.

**Limitations:**

Limitations are discussed in the Appendix.

---

> ### Author Rebuttal · Authors · 2024-08-07
>
> Thank you for your constructive comments. We have taken great care to address all your concerns as follows:
>
> **Comment1(weakness(1) or question(1))： What's the difference between the proposed method and re-weighting method? Why could the proposed method achieve better results?**
>
> **Answer1:**
>
> The mentioned baseline (re-weighting) has been applied in Fair-ARD, while our ABSLD is a re-temperating method. We argue that the re-weighting and re-temperating methods belong to different ideology adopted to seek robust fairness. **The re-temperating method is more direct and accurate than the re-weighting method.**
>
> Specifically, in the optimization process, the essential optimization goal is to reduce the loss between the model's predictions and labels. Re-temperating directly adjust the labels, and its effect can be directly and accurately reflected in the final optimization results of the model. While re-weighting adjusts the loss proportion for different classes, which indirectly affects the model's optimization goal.
>
> **In addition**, we think **re-weighting and re-temperating will not conflict with each other.** Here we try to combine re-weighting and re-temperating strategies. As the result of Table 11 in the overall response PDF, we find that this combination can achieve better robust fairness compared with the re-temperating strategy, which demonstrates that **these two approaches will mutually promote the improvement of robust fairness.**
>
> **Comment2 (weakness(2) or question(3))： Though the proposed method improves the worst class robustness, the overall robustness can't be maintained.**
>
> **Answer2:**
>
> Actually, due to the “buckets effect”, the security of a system often depends on the security of the weakest component. In order to maximize the completion of the model's shortcomings, we focus on improving the model's worst-class robustness. Although the overall robustness remains unchanged or slightly decreases, **our ABSLD obtain the highest robust fairness compared with other methods.** Meanwhile, ABSLD shows the best comprehensive performance of fairness and robustness (NSD) compared with other methods, which means **obtaining the highest robust fairness with sacrificing the least average robustness.**
>
> **Comment3(weakness(3) or question(2))：Question toawrds Eq.(18) and Eq.(19)**
>
> **Answer3:**
>
> We sincerely appreciate your careful and professional check towards Theorem 1. We apologize that some writing mistakes exists in Eq.(19). Actually, from the Eq.(18), we can obtain the above conclusion combined with Eq.(14):
> $$
> \mathbb{E}(\frac{\partial KL(f(x_{c-};\theta_{I}),P_{\lambda1})}{\partial z_{c-}(x_{c-})}) =\mathbb{E}(\frac{\partial KL(f(x_{c+};\theta_{I}),P_{\lambda1})}{\partial z_{c+}(x_{c+})})=\mathbb{E}(p_{c}^{I}(x_{c}) - p_{c-}^{\lambda1}(x_{c-}))=\mathbb{E}(p_{c}^{I}(x_{c}) - p_{c+}^{\lambda1}(x_{c+})),
> $$
> $$
> \mathbb{E}(\frac{\partial KL(f(x_{c+};\theta_{I}),P_{\lambda1})}{\partial z_{c-}(x_{c+})}) =\mathbb{E}(\frac{\partial KL(f(x_{c-};\theta_{I}),P_{\lambda1})}{\partial z_{c+}(x_{c-})})=\mathbb{E}(p_{c}^{I}(x_{c}) - p_{c-}^{\lambda1}(x_{c+}))=\mathbb{E}(p_{c}^{I}(x_{c}) - p_{c+}^{\lambda1}(x_{c-})).
> $$
> And the revised conclusion will not have more negative impacts on subsequent theoretical proofs.
>
> **Comment4(weakness(4) or question(2))：Unclear description about the assumption.**
>
> **Answer4:**
>
> We apologize for the unclear description of this assumption. Here we further clarify it. Actually, the entire assumption is that “If the model is a uniform distribution before the optimization process and the easy class error risk is less than the hard class error risk after optimization process, then the gradient expectation of partial derivative about the easy class’s optimization goal toward the model parameter  is higher than the gradient expectation of partial derivative about the hard class’s optimization goal toward the model parameter”.
>
> **The mathematical explanation for this assumption is as follows:**
>
> In the initial state, we assume that the model is a uniform distribution, and in this case, the error optimization risk is the same for the easy and hard classes:
> $$
> \mathbb{E}(KL(f(x_{c-};\theta_{I}),P_{\lambda1}))=\mathbb{E}(KL(f(x_{c+};\theta_{I}),P_{\lambda1})),
> $$
> then we will simplify the optimization of the model into a one-step gradient iteration process, which means we only consider the initial state before optimization and the last state after optimization. The model parameter is updated from initial $\theta_{I}$ to optimized $\theta_{opt}$. Since easy classes perform better than hard classes after the optimization process, the easy class error risk is smaller than hard class error risk:
> $$
> \mathbb{E}(KL(f(x_{c-};\theta_{opt}),P_{\lambda1}))>\mathbb{E}(KL(f(x_{c+};\theta_{opt}),P_{\lambda1})),
> $$
> the above result means the gradient expectation of partial derivative about the easy class’s optimization goal toward the model parameter ($\mathbb{E}(\frac{\partial KL(f(x_{c-};\theta_{I}),P_{\lambda1})}{\partial \theta})$) is higher than the gradient expectation of partial derivative about the hard class’s optimization goal toward the model parameter ($\mathbb{E}(\frac{\partial KL(f(x_{c+};\theta_{I}),P_{\lambda1})}{\partial \theta})$):
>  $$
>  \mathbb{E}(\frac{\partial KL(f(x_{c-};\theta_{I}),P_{\lambda1})}{\partial \theta})>\mathbb{E}(\frac{\partial KL(f(x_{c+};\theta_{I}),P_{\lambda1})}{\partial \theta}),
>  $$
> so we can obtain the above assumption.

---

> > ### Comment · Reviewer_Dt8R · 2024-08-09
> > **Thanks for the responses from the authors**
> >
> > Thanks for the feedback from the authors. Some problems still make me confused.
> >
> > (1) The proposed algorithm is to allow stronger supervision for hard classes with a smaller temperature during adversarial knowledge distillation. It seems to have similar effects to re-weighting.
> >
> > The re-temperating method adjusts the strength of optimization by assigning different temperatures while the re-weighting can directly adjust the weights for each class.  Could the authors provide any theoretical analysis to show the differences in nature between them? I don't see any advantages of re-temperating over re-weighting.
> >
> > (2) Fairness is important but the overall robustness is also important.
> >
> > (3) I still can't follow the logic that the larger the loss value the larger the gradients in Answer 4. Could you show the detailed derivatives between the loss value and the gradients?

---

> > > ### Author Response · Authors · 2024-08-11
> > >
> > > Thank you again for your valuable comments.
> > >
> > > **Comment5： The proposed algorithm is to allow stronger supervision for hard classes with a smaller temperature during adversarial knowledge distillation. It seems to have similar effects to re-weighting. The re-temperating method adjusts the strength of optimization by assigning different temperatures while the re-weighting can directly adjust the weights for each class. Could the authors provide any theoretical analysis to show the differences in nature between them? I don't see any advantages of re-temperating over re-weighting.**
> > >
> > > **Answer5:**
> > >
> > > Here we re-think and re-explain the advantages of our method and hope for your better understanding:
> > >
> > > Although both methods can play the role of adjusting the optimization strength, the obvious difference exists: **The re-temperating method is directly used to solve the fairness problem by avoiding overfitting of easy class and underfitting of hard class, while the re-weighting method can only alleviate overfitting or underfitting by adjusting the optimization strength, but cannot fundamentally avoid it.**
> > >
> > > Specifically, the fairness problem is essentially that the model overfits the easy class and underfits the hard class, leading to the low error risk for the easy class, and the high error risk for the hard class. Here we formulate the loss function without fairness constraints as follows:
> > > $$
> > > L(x^{adv},x;f _ s,f _ {t}) = \frac{1}{C _ {hard}} \sum_{j=1}^{C _ {hard}} \underbrace{KL(f_{s}(x_{j}^{adv}), f _ {t}({x} _ {j}))} _ {\text{underfitting to hard class}}+\frac{1}{C _ {easy}} \sum_{i=1}^{C _ {easy}} \underbrace{KL(f _ s(x _ i^{adv}), f _ {t}({x} _ i))} _ {\text{overfitting to easy class}},
> > > $$
> > >
> > >
> > > for example, as shown in Figures 6 and 7 in the paper, without additional constraints, the error risk gap between different classes increases as the optimization process continues, which means the error risk of easy class will decrease more and more compared to the error risk of hard class. The results demonstrate that overfitting to easy classes and underfitting to hard classes definitely happen and have a direct relationship with the fairness problem.
> > >
> > >
> > > The re-weighting method only controls the optimization strength via assigning different weights of different classes, generally speaking, the weights $w_j$ of the hard class are larger than weights $w_i$ of the easy class. However, since the final optimization term $KL(f_s(x^{adv}), f_{t}({x}))$ has not changed, it just alleviates but does not solve the phenomenon of overfitting easy classes and underfitting hard classes and cannot fundamentally solve the fairness problem. We formulate the loss function with re-weighting constraints as follows:
> > > $$
> > > L _ {re-weight}(x^{adv},x;f _ s,f _ {t}) = \frac{1}{C _ {hard}} \sum _ {j=1}^{C _ {hard}} w _ j * \underbrace{KL(f _ s(x _ j^{adv}), f _ {t}({x} _ j))} _ {\text{underfitting to hard class}}+\frac{1}{C_{easy}} \sum_{i=1}^{C _ {easy}} w _ i *\underbrace{KL(f _ s(x _ i^{adv}), f _ {t}({x} _ i))} _ {\text{overfitting to easy class}},
> > > $$
> > >
> > > different from the re-weighting method, the re-temperating method directly changes the label smoothness degree via the larger temperature $\tau_{i}^t$ for easy class and smaller temperature $\tau_{j}^t$ for hard class to design a new optimization term $KL(f_s(x^{adv};\tau^s), f_{t}^{'}({x};\tau^t))$: we increase the smoothness degree for easy classes to alleviate the overfitting, and reduce the smoothness degree for hard classes to alleviate the underfitting. In this way, we can achieve a relatively normal-fitting for both easy and hard classes and fundamentally solve the fairness problem. We formulate the loss function with re-temperating constraints as follows:
> > > $$
> > > L _ {re-temperate}(x^{adv},x;f _ s,f _ {t}^{'}) = \frac{1}{C _ {hard}} \sum _ {j=1}^{C _ {hard}}  \underbrace{KL(f _ s(x _ j^{adv};\tau^s), f _ {t}^{'}({x} _ j;\tau _ {j}^t))} _ {\text{normal-fitting to hard class}}+\frac{1}{C _ {easy}} \sum_{i=1}^{C _ {easy}} \underbrace{KL(f _ s(x_i^{adv};\tau^s), f _ {t}^{'}({x} _ i;\tau_{i}^t))} _ {\text{normal-fitting to easy class}}.
> > > $$

---

> > > > ### Comment · Reviewer_Dt8R · 2024-08-12
> > > > **Thanks for the responses from the authors**
> > > >
> > > > I know the differences between re-weighting and re-temperating regarding their implementation. The most important thing is why the re-temperating is better than re-weighting.
> > > > The conclusion is a little bit wired to me that re-temperating can fundamentally solve the fairness problem while re-weighting just alleviates the problem. There should be rigorous analysis.

---

> > > ### Author Response · Authors · 2024-08-11
> > >
> > > **Comment6： Fairness is important but the overall robustness is also important.**
> > >
> > > **Answer6:**
> > > Here we also believe that both overall robustness and fairness are important, and we try to improve fairness while maintaining the overall robustness as much as possible. Although our method obtains the highest robust fairness with sacrificing the least overall robustness and we can achieve the best performance of Normalized Standard Deviation(NSD), e.g., ABSLD reduces the NSD by 0.028, 0.032, 0.017, and 0.024 compared with the best baseline method against the FGSM, PGD, CW, and AA for ResNet-18 on CIFAR-10 (Table 1 in the paper), this trade-off phenomenon does exist similar to other fairness researches [1], [2], [3], [4]. We believe how to solve the trade-off between overall robustness and fairness is one of the directions that should be further explored in the future.
> > >
> > > 1.Xu, H., Liu, X., Li, Y., Jain, A., Tang, J.: To be robust or to be fair: Towards fairness inadversarial training. ICML (2021).
> > >
> > > 2.Ma, X., Wang, Z., Liu, W.: On the tradeoff between robustness and fairness. NeurIPS (2022).
> > >
> > > 3.Li, B., Liu, W.: Wat: improve the worst-class robustness in adversarial training. AAAI (2023).
> > >
> > > 4.Zhang, Y., Zhang, T., Mu, R., Huang, X., & Ruan, W.: Towards Fairness-Aware Adversarial Learning. CVPR (2024).
> > >
> > > **Comment7: I still can't follow the logic that the larger the loss value the larger the gradients in Answer 4. Could you show the detailed derivatives between the loss value and the gradients?**
> > >
> > > **Answer7:**
> > >
> > > **Here we further explain the detailed derivatives as follows:**
> > >
> > > In the initial state, we assume that the model is a uniform distribution, and in this case, the error optimization risk is the same for the easy and hard classes:
> > > $$
> > > \mathbb{E}(KL(f(x_{c-};\theta_{I}),P_{\lambda1}))=\mathbb{E}(KL(f(x_{c+};\theta_{I}),P_{\lambda1})),
> > > $$
> > >
> > > then we will simplify the optimization of the model into a one-step gradient iteration process, which means we only consider the initial state before optimization and the last state after optimization. The model parameter is updated from initial $\theta_{I}$ to optimized $\theta_{opt}$, since easy classes perform better than hard classes after the optimization process, the easy class error risk is smaller than hard class error risk:
> > > $$
> > > \mathbb{E}(KL(f(x_{c-};\theta_{opt}),P_{\lambda1}))>\mathbb{E}(KL(f(x_{c+};\theta_{opt}),P_{\lambda1})),
> > > $$
> > >
> > > at this time, we assume that model's parameter $\theta$ is differentiable and continuous, then we can approximate the gradient expectation of the partial derivative $\mathbb{E}(\frac{\partial KL(f(x_{c-};\theta_{I}),P_{\lambda1})}{\partial \theta})$ and $\mathbb{E}(\frac{\partial KL(f(x_{c+};\theta_{I}),P_{\lambda1})}{\partial \theta})$ as follows:
> > > $$
> > > \mathbb{E}(\frac{\partial KL(f(x_{c-};\theta_{I}),P_{\lambda1})}{\partial \theta})\approx\mathbb{E}(\frac{KL(f(x_{c-};\theta_{opt}),P_{\lambda1})-KL(f(x_{c-};\theta_{I}),P_{\lambda1})}{\theta_{opt}-\theta_{I}}),
> > > $$
> > >
> > > $$
> > > \mathbb{E}(\frac{\partial KL(f(x_{c+};\theta_{I}),P_{\lambda1})}{\partial \theta})\approx\mathbb{E}(\frac{KL(f(x_{c+};\theta_{opt}),P_{\lambda1})-KL(f(x_{c+};\theta_{I}),P_{\lambda1})}{\theta_{opt}-\theta_{I}}),
> > > $$
> > >
> > > combined with the relationship between easy class error risk and hard class error risk, we can obtain the result that the gradient value expectation of partial derivative about the easy class’s optimization goal toward the model parameter ($\mathbb{E}(|\frac{\partial KL(f(x_{c-};\theta_{I}),P_{\lambda1})}{\partial \theta}|)$) is higher than the gradient value expectation of partial derivative about the hard class’s optimization goal toward the model parameter ($\mathbb{E}(|\frac{\partial KL(f(x_{c+};\theta_{I}),P_{\lambda1})}{\partial \theta}|)$) as follows:
> > > $$
> > > \mathbb{E}(|\frac{\partial KL(f(x_{c-};\theta_{I}),P_{\lambda1})}{\partial \theta}|)>\mathbb{E}(|\frac{\partial KL(f(x_{c+};\theta_{I}),P_{\lambda1})}{\partial \theta}|),
> > > $$
> > >
> > > so we can obtain the above assumption.

---

> > > > ### Comment · Reviewer_Dt8R · 2024-08-12
> > > >
> > > > Thanks for the responses from the authors.
> > > >
> > > > It seems that the authors try to bridge the loss value and the gradients with Taylor's Formula.
> > > > Apply Taylor's Formula to the function f(x), we get that f(x) = f(0) + x $f^{'}(x)$.
> > > >
> > > > However, it is still not obvious to draw the conclusion that E($f^{'}(x)$) > E($f^{'}(y)$) when E(f(x)) > E(f(y)).

---

> ### Author Response · Authors · 2024-08-12
>
> Thank you again for your valuable comments.
>
> **Comment8:The most important thing is why the re-temperating is better than re-weighting. The conclusion is a little bit wired to me that re-temperating can fundamentally solve the fairness problem while re-weighting just alleviates the problem. There should be rigorous analysis.**
>
> **Answer8:**
>
> For the study of soft labels, the researcher usually explores and explains their effectiveness from an experimental perspective [1], [2]. Following the previous study, **to demonstrate that our method can suppress the overfitting of the easy class**, we directly compare the class-wise error risk gap between the train set and test set for the re-weighting method (Fair-RSLAD) and the re-temperating method (ABSLD) on easy class (1-th, 2-th, 9-th, 10-th class). The larger error risk gap denotes the less serious the overfitting. The experimental settings of Fair-RSLAD and ABSLD methods are exactly the same except for the re-weight and re-temperate operations. The error risk is obtained based on Cross Entropy Loss. Since the risk change curve cannot be displayed in the form of a picture, we select several typical checkpoints on CIFAR-10 of ResNet-18 in different training periods (the checkpoint in the 200-th, 250-th, and 300-th training epoch). The results are shown in the following Table-A8-1: It can be seen that compared with the re-weighting method, the re-temperating method has a smaller error risk gap between the train set and test set for all the classes, which shows that re-temperating can bring better generalization and can effectively suppress overfitting of easy class.
>
> Meanwhile, **to demonstrate that our method can suppress underfitting of the hard class**, we also list the error risk in the test set (Table-A8-2) for the hard class (3-th, 4-th, 5-th class), the less class-wise error risk denotes the less serious the underfitting. The results denote that the re-temperating method has a smaller error risk in the test set for hard classes, which shows that re-temperating can effectively suppress the underfitting of the hard class.
>
> **So the results directly and effectively illustrate the correctness of our explanation at the experimental level.**
>
> **Table-A8-1: the error risk gap beween the train set and test set for easy class.**
> |method|checkpoint|Class1|Class2|Class9|Class10|
> |:-----:|:----:|:------:|:----:|:----:|:----:|
> |reweight(Fair-RSLAD)|200-th|0.2812|0.2756|0.3317|0.3162|
> |**retemperate(ABSLD)**|200-th|**0.1061**|**0.1075**|**0.1414**|**0.1359**|
> |reweight(Fair-RSLAD)|250-th|0.4252|0.3971|0.4401|0.4481|
> |**retemperate(ABSLD)**|250-th|**0.1560**|**0.1587**|**0.1971**|**0.1914**|
> |reweight(Fair-RSLAD)|300-th|0.4559|0.4174|0.4326|0.4553|
> |**retemperate(ABSLD)**|300-th|**0.1691**|**0.1654**|**0.2012**|**0.1937**|
>
> **Table-A8-2: the error risk in test set for hard class.**
> |method|checkpoint|Class3|Class4|Class5|
> |:-----:|:-----:|:----:|:----:|:----:|
> |reweight(Fair-RSLAD)|200-th|1.571|1.861|1.503|
> |**retemperate(ABSLD)**|200-th|**1.511**|**1.734**|**1.394**|
> |reweight(Fair-RSLAD)|250-th|1.485|1.814|1.358|
> |**retemperate(ABSLD)**|250-th|**1.480**|**1.658**|**1.324**|
> |reweight(Fair-RSLAD)|300-th|1.498|1.780|1.455|
> |**retemperate(ABSLD)**|300-th|**1.483**|**1.613**|**1.360**|
>
> From a theoretical analysis, soft label methods can be applied to reduce the overfitting of unnecessary information by introducing smoothness degree for one-hot labels. **We believe that the re-temperating method fully utilizes the superiority of soft labels and reasonably decouples the superiority from smoothing the labels of the entire data to smoothing the labels of each class.**
>
> Under the premise of effective model capabilities, **the re-tempering method improves the information quality that the model needs to learn, leading to better performance for the model**: for the easy class, model can easily overfit the noise in the training data, so re-temperating introduce more uncertainty or noise in label level by increasing the smoothness degree to avoid the above overfitting; for the hard class, due to the relatively complex characteristics of its samples, the model tends to converge slowly and underfit, so re-temperating introduce less uncertainty or noise in label level by reducing the smoothness degree and give the model a clearer supervision signal to promote its learning of key semantic information.
>
> **The re-weighting method is more like a process of redistributing knowledge information based on class performance, but information quality does not change in this process**, and the model may still overfit some noise in the training data for easy class and underfit some key semantic information for the hard class.
>
> 1.Szegedy, C., Vanhoucke, V., Ioffe, S., Shlens, J. and Wojna, Z.. Rethinking the inception architecture for computer vision. CVPR(2016).
>
> 2.Müller, Rafael, Simon Kornblith, and Geoffrey E. Hinton. "When does label smoothing help?." NeurIPS(2019).

---

> ### Author Response · Authors · 2024-08-12
>
> Thank you again for your valuable comments.
>
> **Comment9: It seems that the authors try to bridge the loss value and the gradients with Taylor's Formula. Apply Taylor's Formula to the function $f(x)$, we get that $f(x) = f(0) + xf^{'}(x)$. However, it is still not obvious to draw the conclusion that $\mathbb{E}(f^{'}(x))>\mathbb{E}(f^{'}(y))$ when $\mathbb{E}(f(x))>\mathbb{E}(f(y))$.**
>
> **Answer9:**
>
> Just as you say, we derive our results similar to the Taylor's Formula $f^{'}(x)=\frac{f(x)-f(0)}{x}$. We definely can not get the result $\mathbb{E}(f^{'}(x))>\mathbb{E}(f^{'}(y))$ when $\mathbb{E}(f(x))>\mathbb{E}(f(y))$. However, we can obtain the $\mathbb{E}(|f(x)-f(0)|)>\mathbb{E}(|f(y)-f(0)|)$ if  $\mathbb{E}(f(0)) >\mathbb{E}(f(y))>\mathbb{E}(f(x))>0$, then we can further obtain $\mathbb{E}(|f^{'}(x)|)>\mathbb{E}(|f^{'}(y)|)$.
>
> **Here we further explain the detailed derivatives and correct the error derivation as follows:**
>
> In the initial state, we assume that the model is a uniform distribution, and in this case, the error optimization risk is the same for the easy and hard classes:
> $$
> \mathbb{E}(KL(f(x_{c-};\theta_{I}),P_{\lambda1}))=\mathbb{E}(KL(f(x_{c+};\theta_{I}),P_{\lambda1})),
> $$
>
> then we will simplify the optimization of the model into a one-step gradient iteration process, which means we only consider the initial state before optimization and the last state after optimization. The model parameter is updated from initial $\theta_{I}$ to optimized $\theta_{opt}$, since easy classes perform better than hard classes after the optimization process, the easy class error risk is smaller than hard class error risk:
> $$
> 0<\mathbb{E}(KL(f(x_{c-};\theta_{opt}),P_{\lambda1}))<\mathbb{E}(KL(f(x_{c+};\theta_{opt}),P_{\lambda1}))<\mathbb{E}(KL(f(x_{c-};\theta_{I}),P_{\lambda1}))=\mathbb{E}(KL(f(x_{c+};\theta_{I}),P_{\lambda1})),
> $$
>
> based on the above result, we can have the result as follows:
> $$
> \mathbb{E}(|KL(f(x_{c-};\theta_{opt}),P_{\lambda1})-KL(f(x_{c-};\theta_{I}),P_{\lambda1})|) > \mathbb{E}(|KL(f(x_{c+};\theta_{opt}),P_{\lambda1})-KL(f(x_{c+};\theta_{I}),P_{\lambda1})|)
> $$
>
> at this time, we assume that model's parameter $\theta$ is differentiable and continuous, then we can approximate the gradient expectation of the partial derivative $\mathbb{E}(\frac{\partial KL(f(x_{c-};\theta_{I}),P_{\lambda1})}{\partial \theta})$ and $\mathbb{E}(\frac{\partial KL(f(x_{c+};\theta_{I}),P_{\lambda1})}{\partial \theta})$ as follows:
> $$
> \mathbb{E}(\frac{\partial KL(f(x_{c-};\theta_{I}),P_{\lambda1})}{\partial \theta})\approx\mathbb{E}(\frac{KL(f(x_{c-};\theta_{opt}),P_{\lambda1})-KL(f(x_{c-};\theta_{I}),P_{\lambda1})}{\theta_{opt}-\theta_{I}}),
> $$
>
> $$
> \mathbb{E}(\frac{\partial KL(f(x_{c+};\theta_{I}),P_{\lambda1})}{\partial \theta})\approx\mathbb{E}(\frac{KL(f(x_{c+};\theta_{opt}),P_{\lambda1})-KL(f(x_{c+};\theta_{I}),P_{\lambda1})}{\theta_{opt}-\theta_{I}}),
> $$
>
> combined with the relationship between easy class error risk and hard class error risk, we can obtain the result that the **gradient absolute value expectation** of partial derivative about the easy class’s optimization goal toward the model parameter ($\mathbb{E}(|\frac{\partial KL(f(x_{c-};\theta_{I}),P_{\lambda1})}{\partial \theta}|)$) is higher than the **gradient absolute value expectation** of partial derivative about the hard class’s optimization goal toward the model parameter ($\mathbb{E}(|\frac{\partial KL(f(x_{c+};\theta_{I}),P_{\lambda1})}{\partial \theta}|)$) as follows:
> $$
> \mathbb{E}(|\frac{\partial KL(f(x_{c-};\theta_{I}),P_{\lambda1})}{\partial \theta}|)>\mathbb{E}(|\frac{\partial KL(f(x_{c+};\theta_{I}),P_{\lambda1})}{\partial \theta}|),
> $$
>
> so we can obtain the above assumption. Actually, the **gradient absolute value expectation** of partial derivative about the class’s optimization goal toward the model parameter denotes the difficulty of the model learning this type of class: the easier the class, the larger the gradient absolute value expectation.

---

> > ### Comment · Reviewer_Dt8R · 2024-08-13
> >
> > Thanks for your responses.
> >
> > For Q8, although your experiments show the advantages of re-temperating over re-weighting. But why is it? Where do the benefits come from?
> >
> > For Q9, it seems that whether the conclusion is established also depends on the sign of ($\theta\_{opt} - \theta\_{I}$)
> >
> > Based on the authors's responses, I will keep my initial rating.

---

> > > ### Author Response · Authors · 2024-08-13
> > >
> > > Thank you again for your valuable comments.
> > >
> > > **Comment10: For Q8, although your experiments show the advantages of re-temperating over re-weighting. But why is it? Where do the benefits come from?**
> > >
> > > **Answer10:**
> > >
> > > Actually, we have explained it in the Q8, the explanation is as follows:
> > >
> > > ''From a theoretical analysis, soft label methods can be applied to reduce the overfitting of unnecessary information by introducing smoothness degree for one-hot labels. **We believe that the re-temperating method fully utilizes the superiority of soft labels and reasonably decouples the superiority from smoothing the labels of the entire data to smoothing the labels of each class.**
> > >
> > > Under the premise of effective model capabilities, **the re-tempering method improves the information quality that the model needs to learn, leading to better performance for the model**: for the easy class, model can easily overfit the noise in the training data, so re-temperating introduce more uncertainty or noise in label level by increasing the smoothness degree to avoid the above overfitting; for the hard class, due to the relatively complex characteristics of its samples, the model tends to converge slowly and underfit, so re-temperating introduce less uncertainty or noise in label level by reducing the smoothness degree and give the model a clearer supervision signal to promote its learning of key semantic information.
> > >
> > > **The re-weighting method is more like a process of redistributing knowledge information based on class performance, but information quality does not change in this process**, and the model may still overfit some noise in the training data for easy class and underfit some key semantic information for the hard class.''
> > >
> > > **Comment11: it seems that whether the conclusion is established also depends on the sign of $\theta_{opt}-\theta_{I}$**
> > >
> > > **Answer11:**
> > >
> > > Since we have added an **absolute value to the value** of this item, its sign of $\theta_{opt}-\theta_{I}$ will not have a final impact on the result:
> > > $$
> > > \mathbb{E}(|\frac{\partial KL(f(x_{c-};\theta_{I}),P_{\lambda1})}{\partial \theta}|)>\mathbb{E}(|\frac{\partial KL(f(x_{c+};\theta_{I}),P_{\lambda1})}{\partial \theta}|),
> > > $$
> > > Actually, we apply the above conclusion about the absolute value for the later derivative. Therefore, it does not affect any subsequent conclusions.

---

### Official Review · Reviewer_Vte2 · 2024-07-10

**Soundness:** 3
**Presentation:** 3
**Contribution:** 2
**Rating:** 5
**Confidence:** 5

**Summary:**

The paper introduces Anti-Bias Soft Label Distillation (ABSLD), a method aimed at improving robust fairness in deep neural networks. This paper identifies the smoothness degree of soft labels as a critical factor influencing this imbalance. ABSLD mitigates the robust fairness problem by adjusting the class-wise smoothness degree of soft labels during the knowledge distillation process. By assigning sharper soft labels to harder classes and smoother ones to easier classes, ABSLD reduces the error risk gap between classes. Extensive experiments on datasets like CIFAR-10 and CIFAR-100 demonstrate that ABSLD outperforms state-of-the-art methods in achieving both robustness and fairness.

**Strengths:**

1. The proposed ABSLD method is innovative and provides a fresh perspective by focusing on the smoothness degree of soft labels, differing from the existing re-weighting approaches.
2. The paper provides a theoretical analysis to support the proposed method, strengthening the validity of the claims.
3. Extensive experiments on different datasets and models demonstrate the effectiveness of ABSLD.

**Weaknesses:**

1. The paper provides many implementation details, but some aspects, such as the choice of hyperparameters (e.g., learning rate, temperature adjustments), could be discussed more thoroughly. Clarifying these details can help replicate the results and understand the method's practical implications.

2. Lack of more related works and SOTA methods, e.g.[1][2].

3. What is the advantage of using Knowledge Distillation (KD)? The baselines used in the paper do not seem strong enough, and current state-of-the-art approaches [1][2] appear to offer better performance. Additionally, employing a teacher model increases the practical costs of time and memory.

4. How about re-weighting + re-temperating? or using re-temperating directly via label smoothing without a teacher model.

5. I am concerned that the comparison between ABSLD and Fair-ARD is due to the hyper-parameter used.


[1] WAT: improve the worst-class robustness in adversarial training, AAAI 2023.

[2] Towards Fairness-Aware Adversarial Learning, CVPR 2024

**Questions:**

See the weaknesses above.

---

> ### Author Rebuttal · Authors · 2024-08-07
>
> Thank you for your constructive comments. We have taken great care to address all your concerns as follows:
>
> **Comment1: The choice of hyperparameters (e.g., learning rate, temperature adjustments)**
>
> **Answer1:**
>
> Following your suggestion, we further discuss the hyper-parameter selection of the initial temperature learning rate $\beta$ and the initial teacher’s temperature $\tau_{k}^{t}$ on CIFAR-10 of ResNet-18. As Tables 9 and 10 in the overall response PDF show, the value of $\beta$ and $\tau_{k}^{t}$ can slightly influence the final results in a proper range, and the selection of hyper-parameter (initial $\beta$ as 0.1 and initial $\tau_{k}^{t}$ as 1) is reasonable.
>
> **Comment2: Lack of more related works and SOTA methods, e.g.[1][2].**
>
> **Answer2:**
>
> Following your suggestion, we compare our ABSLD with those two methods (WAT[1] and FAAL[2]). WAT[1] and FAAL are the re-weighting methods, which are different from our ABSLD (re-temperating method). Since FAAL does not provide reproducible open-source code, we directly apply the experimental results on CIFAR-10 of PRN-18 in FAAL’s origin paper. Following FAAL, we add the EMA operation to maintain consistency in the experimental setup. As the following table shows, our method outperforms FAAL by 1.02% and 0.2% in the average robustness and worst-class robustness under AA attack, which demonstrates the effectiveness.
>
> | Method  | AA Avg. (%) | AA Worst (%) |
> | :----: | :----: | :----: |
> | WAT  | 46.16 | 30.70 |
> |FAAL | 49.10 | 33.70 |
> | **ABSLD(ours)** | **50.12** | **33.90** |
>
> **Comment3: What is the advantage of using Knowledge Distillation (KD)? The baselines used in the paper do not seem strong enough, and current state-of-the-art approaches [1][2] appear to offer better performance. Additionally, employing a teacher model increases the practical costs of time and memory.**
>
> **Answer3:**
>
> **Firstly**, We think that applying KD has advantages as follows:
>
> 1. **Adversarial Robustness Distillation (ARD) is a type of state-of-the-art adversarial training method currently.** ARD can effectively bring strong adversarial robustness for trained models within the framework of KD. Therefore, based on the impressive performance, we can maximally pursue approaches that bring both strong overall robustness and fairness just as in previous work [3].
>
> 2. **Knowledge Distillation itself can bring competitive robust fairness.** We can notice that KD-based methods themselves (e.g., RSLAD and AdaAD) have competitive robust fairness compared with the other baseline methods (e.g., TRADES, FRL, and CFA). We believe that the KD-based method is more friendly in improving the robust fairness of the model. So we select the Knowledge Distillation as the baseline.
>
> **Secondly**, the results in **Answer2** demonstrate the effectiveness of our ABSLD compared with WAT [1] and FAAL [2].
>
> **Thirdly**, for ABSLD, the teacher model is only applied to generate the soft labels without updating its parameters, thus, the additional cost is limited. Meanwhile, we calculate the time cost and memory. Here we calculate the costs on CIFAR-10 of ResNet-18. As the following table shows, we think the costs of time and memory can be acceptable while bringing performance improvements compared with main-stream methods.
>
> | Method  | Time（Avg. Epoch）|	GPU Memory
> | :----: | :----: | :----:
> | SAT  | 175s | 2764MiB
> | WAT | 284s | 3624MiB
> | ABSLD(ours)| 224s | 3832MiB |
>
> **Comment4: How about re-weighting + re-temperating? or using re-temperating directly via label smoothing without a teacher model.**
>
> **Answer4:**
>
> Thank you for your valuable comment. Following your suggestion, we try to combine re-weighting and re-temperating strategies and evaluate the adversarial robust fairness. As the results of Table 11 in the overall response PDF, we find that re-weighting + re-temperating can achieve better robust fairness compared with the re-temperating strategy, which demonstrates that **these two approaches will mutually promote the improvement of fairness without conflicting with each other.**
>
> Meanwhile, we also try to explore the performance of using re-temperating directly via label smoothing without a teacher model and add the performance of label smoothing as the baseline for comparison. The results of Table 12 in the overall response PDF show that despite the re-temperating directly via label smoothing is better than the baseline, a certain gap still exists compared to the ARD-based method.
>
> **Comment5: I am concerned that the comparison between ABSLD and Fair-ARD is due to the hyper-parameter used.**
>
> **Answer5:**
>
> For the Fair-ARD, we completely retain the hyper-parameter settings and select the best performance versions shown in Fair-ARD's original paper (Fair-ARD for CIFAR-10 and Fair-RSLAD for CIFAR-100). To further clarify the effectiveness, we compare our ABSLD with Fair-ARD's different version on CIFAR-10 of ResNet-18, including Fair-ARD, Fair-IAD, Fair-MTARD, and Fair-RSLAD, and the results of Table 13 in the overall response PDF show that our ABLSD has better performance compared with different types of Fair-ARD versions.
>
> Especially, for the comparison between Fair-RSLAD and ABSLD, except for the specific hyper-parameters (e.g., the Fair-RSLAD’s for re-weighting or the ABSLD’s for re-temperating), the baseline (RSLAD) and its corresponding hyper-parameters are completely the same. **Thus, the effectiveness of ABSLD compared with Fair-ARD does not come from intentional hyper-parameter selection.**
>
> 1.Wat: improve the worst-class robustness in adversarial training. AAAI(2023).
>
> 2.Towards Fairness-Aware Adversarial Learning. CVPR(2024).
>
> 3.Revisiting adversarial robustness distillation from the perspective of robust fairness. NeurIPS(2023).

---

### Official Review · Reviewer_6NLr · 2024-07-17

**Soundness:** 2
**Presentation:** 3
**Contribution:** 2
**Rating:** 3
**Confidence:** 4

**Summary:**

This paper explores the issue of robust fairness in deep neural networks (DNNs), particularly focusing on the disparity in robustness between different classes in adversarial training (AT) and adversarial robustness distillation (ARD) methods. The authors propose a novel method called Anti-Bias Soft Label Distillation (ABSLD) that aims to mitigate this problem by adjusting the smoothness degree of soft labels for different classes during the knowledge distillation process. ABSLD adaptively reduces the error risk gap between classes by re-tempering the teacher's soft labels with different temperatures, which are determined based on the student's error risk. Extensive experiments demonstrate that ABSLD outperforms existing AT, ARD, and robust fairness methods in terms of a comprehensive metric that combines robustness and fairness, known as the Normalized Standard Deviation. The paper contributes to the literature by providing both empirical observations and theoretical analysis on the impact of soft label smoothness on robust fairness and by advancing a new technique within the knowledge distillation framework to achieve better adversarial robust fairness.

**Strengths:**

1.  Well-designed experiments: used (Tiny)-Imagenet,  set up AA as attack, and leverage NSD as metric

**Weaknesses:**

1.  In my humble opinion, I do not regard adversarial "fairness" is a critical problem. It is different from other fairness problem (e.g. demographic features) which brings social impacts.
2.  For technical contribution, soft-label, knowledge distillation, as well as retemperature are well-known techniques. This paper applies the these techniques on the specific problem, but does not provide specific designs with respect to the problem.
3. According to the experiment results, this method also suffers the trade-off between clean acc and robustness, which indicates not to solve the critical issue with respect to the adversarial training.

**Questions:**

Please help to check weaknesses.

**Limitations:**

Also in weaknesses

---

> ### Author Rebuttal · Authors · 2024-08-07
>
> Thank you for your constructive comments. We have taken great care to address all your concerns as follows:
>
> **Comment1:In my humble opinion, I do not regard adversarial "fairness" is a critical problem. It is different from other fairness problem (e.g. demographic features) which brings social impacts.**
>
> **Answer1:**
>
> Actually, we think **adversarial fairness problem is an issue worthy of being further studied in the field of adversarial robustness.** Previous research has always focused on improving overall robustness, however, the adversarially-trained models may exhibit high robustness in some classes while significantly low robustness in other classes. Due to the “buckets effect”, the security of a system often depends on the security of the weakest component. Specifically, an overall robust model appears to be relatively safe for model users, **however, the robust model with poor robust fairness will lead to attackers targeting vulnerable classes of the model, which leads to significant security risks to potential applications,** e.g., an autonomous driving system that has high robustness for inanimate objects on the road but lacks robustness when detecting pedestrians, may be misled by adversarial examples, leading to traffic accidents [6].
>
>  Due to the importance, many studies have been published to solve this critical issue, e.g., FRL[1] (ICML2021), FAT[2] (NeurIPS2022), BAT[3] (AAAI2023), WAT[4] (AAAI2023), CFA[5] (CVPR2023), Fair-ARD[6] (NeurIPS2023), TAAL[7] (CVPR2024). **We believe that related research is necessary and will make a positive contribution to the secure application of AI.**
>
> **Comment2: For technical contribution, soft-label, knowledge distillation, as well as retemperature are well-known techniques. This paper applies the these techniques on the specific problem, but does not provide specific designs with respect to the problem.**
>
> **Answer2:**
>
> Here we mainly want to clarify the technical contribution of the proposed method.
>
> Previous works always apply the re-weighting ideology to achieve robust fairness for different types of classes, however, as another important factor in the optimization objective function, the role of the labels has been ignored by previous researchers. Inspired by this, we try to explore robust fairness from the perspective of samples' soft labels.
>
> To the best of our knowledge, **we are the first one to explore the labels' effects on the adversarial robust fairness of DNNs**, which is different from the existing sample-based perspective. We find that the smoothness degree of samples' soft labels for different types of classes can affect the robust fairness from both empirical observation and theoretical analysis.
>
> To further enhance adversarial robust fairness, we propose a specific and novel method named ABSLD. Specifically, we re-temperate the teacher's soft labels to adjust the class-wise smoothness degree and further reduce the student's error risk gap between different classes.  The extensive experiments show our ABSLD can outperform other state-of-the-art methods in robust fairness problems. **We sincerely believe that our proposed method is innovative and effective.**
>
> **Comment3: According to the experiment results, this method also suffers the trade-off between clean acc and robustness, which indicates not to solve the critical issue with respect to the adversarial training.**
>
> **Answer3:**
>
> Actually, **ABSLD is proposed to solve the adversarial robust fairness problem, which is distinct from the accuracy-robustness trade-off problem.** We believe that even if the overall robustness of a model is high, the poor robustness on a specific class of data can also pose security issues. In order to maximize the completion of the model's shortcomings, we focus on improving the model's robust fairness problem.
>
> 1.Xu, H., Liu, X., Li, Y., Jain, A., Tang, J.: To be robust or to be fair: Towards fairness in adversarial training. ICML (2021).
>
> 2.Ma, X., Wang, Z., Liu, W.: On the tradeoff between robustness and fairness. NeurIPS (2022).
>
> 3.Sun, C., Xu, C., Yao, C., Liang, S., Wu, Y., Liang, D., Liu, X., Liu, A.: Improving robust fariness via balance adversarial training. AAAI (2023).
>
> 4.Li, B., Liu, W.: Wat: improve the worst-class robustness in adversarial training.AAAI (2023).
>
> 5.Wei, Z., Wang, Y., Guo, Y., Wang, Y.: Cfa: Class-wise calibrated fair adversarial training. CVPR (2023).
>
> 6.Xinli, Y., Mou, N., Qian, W., Lingchen, Z.: Revisiting adversarial robustness distillation from the perspective of robust fairness. NeurIPS (2023).
>
> 7.Zhang, Y., Zhang, T., Mu, R., Huang, X., & Ruan, W.: Towards Fairness-Aware Adversarial Learning. CVPR (2024).

---

### Official Review · Reviewer_H27W · 2024-07-24

**Soundness:** 3
**Presentation:** 3
**Contribution:** 3
**Rating:** 5
**Confidence:** 4

**Summary:**

This paper discusses the robust fairness problem, which is essential to solve for reducing concerns surrounding class-based security. The paper majorly analyzed the inheritance of robust fairness during adversarial robustness distillation (ARD). It is found that student models only partially inherit robust fairness from teacher models. To solve this, authors have examined how the degree of smoothness of samples' soft labels influences class-wise fairness in (ARD). They empirically and theoretically prove that appropriately assigning class-wise smoothness degrees of soft labels during ARD can be beneficial to achieve robust fairness.

Therefore, as a solution to the fairness problem associated with ARD, they propose the Anti-Bias Soft Label Distillation (ABSLD), a knowledge distillation framework designed to reduce error risk gaps between student classes by adjusting the class-specific smoothness degree of teacher's soft labels during training, controlled by assigning temperatures. The authors provided experiments to demonstrate their method's effectiveness.

**Strengths:**

- The empirical and theoretical analysis on the impact of the smoothness degree of soft labels on class-wise fairness is interesting and well documented
-  Within the knowledge distillation framework, the use of temperature as a means of controlling the smoothness of the teacher's soft labels during training has proven to reduce the student's class-wise risk gap. Thus, improving the class-wise fairness.

**Weaknesses:**

- The evaluations on common corruptions is missing. (Does this method transfers robustness and fairness towards common corruptions as well?)
- The evaluations on related fairness works, Wat [1] and FAT[2] is missing.
- Most of the works mentioned as baselines focusing on fairness (CFA, BAT, FRL), in the paper are trained using PRN-18. May be better to evaluate the current approach using the PRN-18 as well (Optional).
- The authors should explore if their method is also fair and robust in ViTs. The architectures suitable for training ARD and AT approaches are available in robustness benchmark [3].
- Few ARD related evaluations  like IAD, MTARD  in combination with Fair-ARD approach as mentioned in Fair-ARD [4] needs to be evaluated.
- Comparison of ABSLD with Fair-ARD in writing can be improved. How different is approaches from each other, what benefits ABSLD the most over Fair-ARD?


[1] Li, B., Liu, W.: Wat: improve the worst-class robustness in adversarial training. In: Proceedings of the AAAI Conference on Artificial Intelligence. vol. 37, pp. 14982–14990 (2023)
[2] Jingfeng Zhang, Xilie Xu, Bo Han, Gang Niu, Lizhen Cui, Masashi Sugiyama, and Mohan Kankanhalli. Attacks which do not kill training make adversarial learning stronger. In International conference on machine learning, pages 11278–11287. PMLR, 2020.
[3] Francesco Croce, Maksym Andriushchenko, Vikash Sehwag, Edoardo Debenedetti, Nicolas Flammarion, Mung Chiang, Prateek Mittal, and Matthias Hein. Robustbench: a standardized adversarial robustness benchmark.
[4] Xinli, Y., Mou, N., Qian, W., Lingchen, Z.: Revisiting adversarial robustness distillation from the perspective of robust fairness. NeurIPS (2023)

**Questions:**

How is the smoothness degree of different classes (easy & hard) during the training?

**Limitations:**

Broader Impact : Potential positive societal impacts and negative societal impacts of the work is missing but its mentioned as included.

---

> ### Author Rebuttal · Authors · 2024-08-07
>
> Thank you for your constructive comments. We have taken great care to address all your concerns as follows:
>
> **Comment1: The evaluations on common corruptions is missing.**
>
> **Answer1:**
>
> We have chosen two common corruptions: Gaussian noise and color channel transformation. As in Table 8 in the overall response PDF, ABSLD improves worst-class robustness by 0.9% and 1.8%, and reduces NSD by 0.012 and 0.01 on CIFAR-10 of ResNet-18 under those two attacks. The results show that ABSLD can transfer robust fairness towards common corruptions.
>
> **Comment2: The evaluations on Wat [1] and FAT[2] is missing.**
>
> **Answer2:**
>
> Here we add the comparison with WAT[1] and FAT[2]. As the following table shows, ABSLD improves worst-class robustness by 0.8% and reduces NSD by 0.03 compared with the second-best method on CIFAR-10 of ResNet-18 under AA attack.
>
> Method  | AA Avg.(%) | AA Worst(%) | AA NSD
> :----: |:----: | :----: | :----:
>  WAT | 46.19 | 30.20 | 0.286
>   FAT | 41.83 | 16.80 | 0.409
> **ABSLD(ours)** | **50.25** | **31.00** | **0.256**
>
> **Comment3: The application of PRN-18.**
>
> **Answer3:**
>
> We select the trained model as ResNet-18 following Fair-ARD [3], which is also the common setting in Adversarial Robustness Distillation (e.g., ARD, IAD, and RSLAD). To further verify the effectiveness, we also train PRN-18 on CIFAR-10 via ABSLD and the results in the following Table demonstrate the effectiveness of ABSLD on PRN-18.
>
> | Method | AA Avg.(%) | AA Worst(%)  | AA NSD
> | :----: |:----: | :----: |  :----:
> | FRL  | 45.90 | 25.40  | -
> | CFA  | 50.03 | 26.50  | 0.301
> | **ABSLD(ours)** | **50.12** | **33.90**  | **0.263**
>
> **Comment4: fairness and robustness in ViTs.**
>
> **Answer4:**
>
> Here we select ViT-B to train following [4]. The result shows that ABSLD achieves better performance compared with the baseline method (RSLAD), so ABSLD is effective applied to ViTs but not limited to CNNs.
>
> | Method | AA Avg.(%) | AA Worst(%) | AA NSD
> | :----: |:----: | :----: | :----:
> | RSLAD  | **49.07** | 17.50 | 0.379
> | **ABSLD(ours)** | 47.73 | **20.70** | **0.327** |
>
> **Comment5:IAD, MTARD in combination with Fair-ARD needs to be evaluated.**
>
> **Answer5:**
>
> Following your suggestion, we evaluate the Fair-ARD, Fair-IAD, Fair-MTARD, and Fair-RSLAD (different versions of Fair-ARD) for comparison with the ABSLD on CIFAR-10 of ResNet-18. As Table 13 in the overall response PDF shows, ABSLD improves worst-class robustness by 5.6% and reduces NSD by 0.049 under AutoAttack, which further demonstrates the effectiveness.
>
> **Comment6: Comparison of ABSLD with Fair-ARD in writing can be improved.**
>
> **Answer6:**
>
> **Fair-ARD exists two major distinctions with our ABSLD as follows:**
>
> 1. From the optimization perspective, ABSLD redesign a new loss function by adjusting **different smoothness degree of soft labels** for different classes. Fair-ARD  modifies the existing loss function by adjusting **different weights** for different classes.
>
> 2. From the designed method, ABSLD applies **the optimization error risk as a metric** to adaptively re-temperate the label smoothness degree for different classes. Fair-ARD applies **the least PGD steps for generating adversarial examples as a metric** to adaptively re-weight for different classes.
>
> **We think that the following advantage benefits ABSLD the most over Fair-ARD:**
>
> We argue that the re-weighting method (e.g., Fair-ARD) and re-temperating method (our ABSLD) belong to different implementation paths to seek robust fairness.  **The re-temperating method is more direct and accurate than the re-weighting method.**
>
> **Specifically**, in the optimization process, the essential optimization goal is to reduce the loss between the model's predictions and labels. Re-temperating directly adjust the labels, and its effect can be directly and accurately reflected in the final optimization results of the model. While re-weighting adjusts the loss proportion for different classes, which indirectly affects the model's optimization goal.
>
> **In addition**, due to different implementation paths, **re-weighting and re-temperating methods will not conflict with each other.** Here we try to combine re-weighting and re-temperating strategies. As the results of Table 11 in the overall response PDF, we find that this combination can achieve better robust fairness compared with the re-temperating strategy, which demonstrates that **these two approaches will mutually promote the improvement of robust fairness.**
>
> **Comment7(question1): The smoothness degree of different classes.**
>
> **Answer7:**
>
> In the training process, **the smoothness degree of the easy class will be smoother, and the smoothness degree of the hard class will be sharper.** In Figure 7 of the overall response PDF, we visualize the trend about the smoothness degree for easy classes (2-th) and hard classes (4-th) on CIFAR-10, which can further confirm our explanation. Here we measure the smoothness degree through information entropy, where the larger entropy denotes the smoother smoothness degree and the smaller entropy denotes the sharper smoothness degree.
>
> 1.Wat: improve the worst-class robustness in adversarial training. AAAI(2023).
>
> 2.Attacks which do not kill training make adversarial learning stronger. ICML(2020).
>
> 3.Revisiting adversarial robustness distillation from the perspective of robust fairness. NeurIPS(2023).
>
> 4.When adversarial training meets vision transformers: Recipes from training to archi tecture. NeurIPS(2022).

---

### Author Rebuttal · Authors · 2024-08-07

This response contains mainly an overall response PDF with details as follows:

Figure 7: Information entropy change curve of teacher soft labels for hard classes and easy classes.

Table 8: Results on two common corruptions, for Gaussian Noise(GN) and Colour Channel Transformations(CCT).

Table 9: Discussion about different Initial $\tau_{k}^{t}$.

Table 10: Discussion about different Initial $\beta$.

Table 11: Comparison between reweighting, retemperating, and their combination.

Table 12: Comparison between Labelsmoothing, re-temperating via Labelsmoothing, and re-temperating via KD (ABSLD).

Table 13: Comparison between ABSLD and different Fair-ARD's versions.

---

### Decision · Program_Chairs · 2024-09-25

**Decision:**

Accept (poster)

**Comment:**

The submitted manuscript addresses the problem of adversarial fairness, i.e. the problem that there is a severe trade-off between adversarial robustness and model fairness. The paper has received four reviews, three of which, after discussion, agree on the merits of the paper. Although the proposed approach recombines existing techniques to improve adversarial fairness, it shows overall promising results on an important topic. This is an overall sound paper which offers a step towards advancing an important field of research, adversarial fairness. This is different from the research question regarding the trade-off between clean and robust accuracy, but equally valuable.